# Influence of reservoir geology on seismic response during decameter scale hydraulic stimulations in crystalline rock

Villiger, Linus[1], Gischig, Valentin, Samuel[2], Doetsch, Joseph[3], Krietsch, Hannes[3], Dutler, Nathan, Oliver[4], Jalali, Mohammadreza[5], Valley, Benoît[4], Selvaduai, Paul, Antony[1], Mignan, Arnaud[1, 6], Plenkers, Katrin[1], Giardini, Domenico[1], Amann, Forian[5], Wiemer, Stefan[1]

[1]Swiss Seismological Service, ETH Zurich, Zurich, Switzerland
[2]CSD Ingenieure, Bern, 3097, Switzerland
[3]Department of Earth Sciences, ETH Zurich, Zurich, Switzerland
[4]CHYN, University of Neuchâtel, Neuchâtel, Switzerland
[5]Department of Engineering Geology & Hydrogeology, RWTH Aachen, Aachen, Germany
[6]Institute of Risk Analysis, Prediction and Management, Academy for Advanced Interdisciplinary Studies, Southern University of Science and Technology, Shenzhen, China

Correspondence to: Linus Villiger (linus.villiger@sed.ethz.ch)

**Abstract.** We performed a series of 12 hydraulic stimulation experiments in a 20 x 20 x 20 m foliated, crystalline rock volume intersected by two distinct fault sets at the Grimsel Test Site, Switzerland. The goal of these experiments was to improve our understanding of stimulation processes associated with high-pressure fluid injection used for reservoir creation in enhanced or engineered geothermal systems. In the first six experiments, pre-existing fractures were stimulated to induce shear dilation and enhance permeability. Two types of shear zones were targeted for these hydroshearing experiments: *i*) ductile ones with intense foliation and *ii*) brittle-ductile ones associated with a fractured zone. The second series of six stimulations were performed in borehole intervals without natural fractures to initiate and propagate hydraulic fractures that connect the wellbore to the existing fracture network. The same injection protocol was used for all experiments within each stimulation series so that the differences observed will give insights into the effect of geology on the seismo-hydro-mechanical response rather than differences due to the injection protocols. Deformations and fluid pressure were monitored using a dense sensor network in boreholes surrounding the injection locations. Seismicity was recorded with sensitive in-situ acoustic emission sensors both in boreholes and at the tunnel walls. We observed high variability in the seismic response in terms of seismogenic indices, b-values, spatial and temporal evolution during both hydroshearing and hydrofracturing experiments, which we attribute to local geological heterogeneities. Seismicity was most pronounced for injections into the highly conductive brittle-ductile shear zones, while injectivity increase on these structures was only marginal. No significant differences between the seismic response of hydroshearing and hydrofracturing was identified, possibly because the hydrofractures interact with the same pre-existing fracture network that is reactivated during the hydroshearing experiments. Fault slip during the hydroshearing experiments was predominantly aseismic. The results of our hydraulic stimulations indicate that stimulation of short borehole intervals with limited fluid volumes (i.e., the concept of zonal insulation) may be an effective approach to limit induced seismic hazard if highly seismogenic structures can be avoided.

## 1 Introduction

*Opportunities and challenges of enhanced geothermal systems (EGS)*

Our global primary energy demand is predicted to increase (McKinsey, 2016;WorldEnergyConcil, 2016), while at the same time we urgently need to de-carbonise our economies. Geothermal energy represents a promising

option, because it taps vast geothermal resources, which is considered to be an almost greenhouse gas emission free primary energy resource (Tester et al., 2006). Geothermal power plants convert geothermal heat to base-load power - an important capability in combination with increasing capacities of "new renewables", for example, solar and wind-based power production, which is dependent on temporal weather conditions. Of particular interest are the so-called enhanced or engineered geothermal systems (EGS), which are less dependent on specific geological site conditions, such as volcanic areas or those with sufficient natural fluid flow, and are being investigated as viable options to the increased alternative energy demand.

EGS allows advective heat extraction from hot formations by pumping a working fluid from injection wells through the fractured formation to one or several production wells. For the crystalline rock often found at depths of 3 to 6 km, where sufficient temperature for an economic electric power production (Evans, 2014) is available, permeability is usually too low for advective heat transport (Ingebritsen & Manning, 2010; Preisig et al., 2015). Therefore, permeability has to be enhanced artificially with high-rate fluid injection (i.e. hydraulic stimulation). Once permeability is sufficiently and permanently increased, fluid flow and heat exchange between the rock and the fluid is possible.

The first efforts towards EGS date back to a project performed at Fenton Hill in the early 1970s (Brown et al., 2012). Since then, multiple projects in research and industry have been performed without reaching technical maturity and economical standards (Jung, 2013a). More recent projects, like the project in Basel, Switzerland in 2006 (Mignan et al., 2015) and Pohang, South Korea in 2017 (Grigoli et al., 2018; Kim et al., 2018; Kang-Kun et al., 2019) have seen major set-backs because of induced earthquakes with unacceptably large magnitudes. One ongoing project that successfully completed the first reservoir stimulation phase is located beneath Helsinki (Kwiatek et al., 2019).

Hydraulic stimulation inevitably leads to induced seismicity, but the large majority of events are not felt; this has been defined as micro-seismicity (Ellsworth, 2013). Micro-seismic clouds are used to trace developing fracture networks and potential fluid flow paths (Bohnhoff et al., 2009;Shapiro, 2015) and represent an important monitoring tool for reservoir characterization during the stimulation process. However, in some instances damaging earthquakes have occurred and pose a threat to local communities and infrastructure (Grigoli et al., 2018; Kim et al., 2018; Mignan et al., 2015). Slightly damaging or felt induced seismicity may have a severe impact on public acceptance of EGS (Rubinstein & Mahani, 2015; Trutnevyte & Wiemer, 2017) and on the financial feasibility of EGS projects (Mignan et al., 2019). Grigoli et al. (2017) have suggested that stimulation processes are technically lacking and need improvement – specifically the complex coupling between the geomechanical and seismic response of the reservoir. Advances in this area might be the key to producing safer and productive EGS power plants.

*Geomechanics of EGS stimulation processes*

The dominant stimulation mechanism in EGS has been identified as the shearing of pre-existing fractures and faults, known as hydraulic shearing (HS), and characterized as mode-II and mode-III or a combination of both dislocations (Fehler, 1989;Kelkar et al., 2016;Pine and Batchelor, 1984). The most common mechanism leading to HS is triggered by high pressure fluid injection. The presence of higher fluid pressure on the fault lowers the effective normal stresses across pre-existing discontinuities, which enables shear-dislocation. The pore pressure increase needs to be above shear strength of pre-existing discontinuities, but may not exceed the minimal principal stress magnitude. A prerequisite for shearing are discontinuities that support a sub-critical level of shear stress.

The dislocation caused by shearing is, to a large extent, irreversible. Fracture plane roughness (i.e. asperities) leads to permanent dilation and thus to increased fracture permeability by up to 2-3 orders of magnitude (Evans et al., 2005; Häring et al., 2008). Due to such self-propping, HS is a key mechanism for permeability creation in EGS reservoirs. While the reduction in effective stress due to pore pressure increases is the primary underlying mechanism, seismicity may also be triggered beyond the volume affected by fluid pressure diffusion (here referred to as the pressurized zone). Possible additional mechanisms may be related to poro-elastic stress transfer (Goebel et al., 2016;Goebel et al., 2017;Goebel and Brodsky, 2018) or slip-related Coulomb stress redistribution (Catalli et al., 2016;Schoenball and Ellsworth, 2017).

Another stimulation mechanism is the formation and propagation of new tensile fractures (also known as hydraulic fracturing, HF, i.e., mode-I opening) in intact rock (Economides & Nolte, 1989). The injection pressure is increased within short borehole intervals in zones with no pre-existing fractures until it exceeds the formation breakdown pressure (Hubbert and Willis, 1957). Newly initiated fractures tend to grow perpendicular to the minimum principal compressional stress component acting on a volume (Haimson & Fairhurst, 1969). The propagation of a HF at propagation pressure is inhibited if it intersects a pre-existing fracture that supports shear stresses high enough for slip to be induced (i.e. HS) at pressures below the ones required for HF (McClure & Horne, 2013). Thus, driving HFs over long distances in a fractured rock mass may be difficult because of interaction with the pre-existing natural fracture network that causes the fluid pressure driving the fracture to “leak-off”. In addition, the decrease in pore pressure with increasing distance possibly restricts HFs to areas near the injection interval. In contrast to sheared fractures, HF close almost reversibly once the fluid pressure is released (Secor Jr & Pollard, 1975). Moreover, due to the geologic complexities of the targeted reservoirs, shearing and fracturing are likely to appear in tandem. Computational modelling approaches of hydraulic stimulation have shown that fracture opening (HF) and shear dislocation (HS) possibly act in combination (McClure and Horne, 2014).

*Mediation of induced seismicity*

An important issue for EGS sites is the a priori assessment of seismic hazard and risk, which typically includes forecasting seismicity rates defined by the seismogenic index (Shapiro et al., 2010) – also called activation feedback a-value (Broccardo et al., 2017; Mignan et al., 2017; Mignan et al., 2019) and the Gutenberg-Richter b-value, as well as the maximum possible earthquake magnitude ($M_{max_pos}$). One approach to estimate the latter is McGarr’s (2014) upper limit for the maximum possible magnitude $M_{\max_McGarr} = \frac{2}{3}\log 10(G \cdot V) - 10.7 + \frac{14}{3}$, where, $V$ and $G$, represent the injected volume and the shear modulus, respectively (Figure 1). This reflects the rather optimistic view that the maximum possible induced earthquake in a reservoir with shear modulus $G$ depends on the injected volume, leading to a volume change and alterations in the deviatoric stresses, which are relaxed by the induced earthquake. However, a number of case studies point out that this threshold is exceeded, for example in Pohang, South Korea (Kang-Kun et al., 2019), St.Gallen, Switzerland (Diehl et al., 2017) and Western Canada (Atkinson et al., 2016). Thus, an alternative and more conservative view, shared by various authors (e.g., Mignan et al. (2015);Mignan et al. (2017);Atkinson et al. (2016);Van der Elst et al. (2016);Gischig (2015)) suggests that the largest possible induced earthquake corresponds to the largest possible natural earthquake defined by the seismotectonic setting at a respective site. However, Figure 1 carries an additional important message: For a given injected volume (e.g. 10’000 m$^3$) $M_{max_obs}$ may reach from $M_w$ -1.0 to $M_w$ > 5.0. It is extremely important to better understand the physical mechanisms responsible for over- and under-estimates of $M_{max_pos}$. In order to advance

EGS technology, we need to understand the causes of the large variability in seismicity across multiple stimulation projects on variable scales and find strategies to promote low levels of seismicity.

Dinske and Shapiro (2013), among others (Mignan et al., 2017), have shown that seismicity rates might be linked to the geologic setting. Gischig (2015) propose that these observations might make the prediction of seismicity rates and $M_{max_pos}$ highly site-specific and greatly influenced by the stress conditions, the proximity to faults and other local geological conditions. Seismicity may also be dependent on the orientation of faults in the prevailing stress field. Gischig (2015) concludes that optimally oriented faults may rupture in an uncontrolled fashion beyond the pressurized volume and stop where geological conditions change. In contrast, rupture along less favorably oriented faults have a larger portion of aseismic slip and this slip arrests within, or only a little beyond, the pressurized volume. The seismic response to fluid injection may also be linked to the local geological setting. McClure and Horne (2014) relate the formation properties observed in the wellbores of six field scale hydraulic stimulations in granitic rock to the severity of the seismic response. They suspect that there is a correlation between fault maturity (i.e., well-developed brittle fault zones) and high seismic moment release. De Barros et al. (2016) also suspect that the seismic behaviour to fluid injection is dependent on the fault damage zone architecture.

The aforementioned studies suggest that the site-specific geological conditions (rock type, mechanical and geometrical properties of faults and fractures, stress conditions, etc.) may control induced seismicity. Nevertheless, some studies also focus on injection strategies that may reduce induced seismicity. Yoon et al. (2014) and Zang et al. (2018) suggest that a fatigue hydraulic fracturing injection scheme, including cyclic injection pressure, may lead to lower fracture breakdown pressures, a systematic reduction of the maximum magnitude and an increased hydraulic performance when compared to conventional monotonic high-pressure fluid injection. Although many alternative injection strategies are widely discussed in the literature (e.g., McClure et al. (2016), Zimmermann et al. (2014), McClure and Horne (2011)), experimental evidence for advantageous injection schemes are difficult to obtain, as it is not clear to what degree geological conditions or the injection protocol are responsible for variable seismicity outcomes.

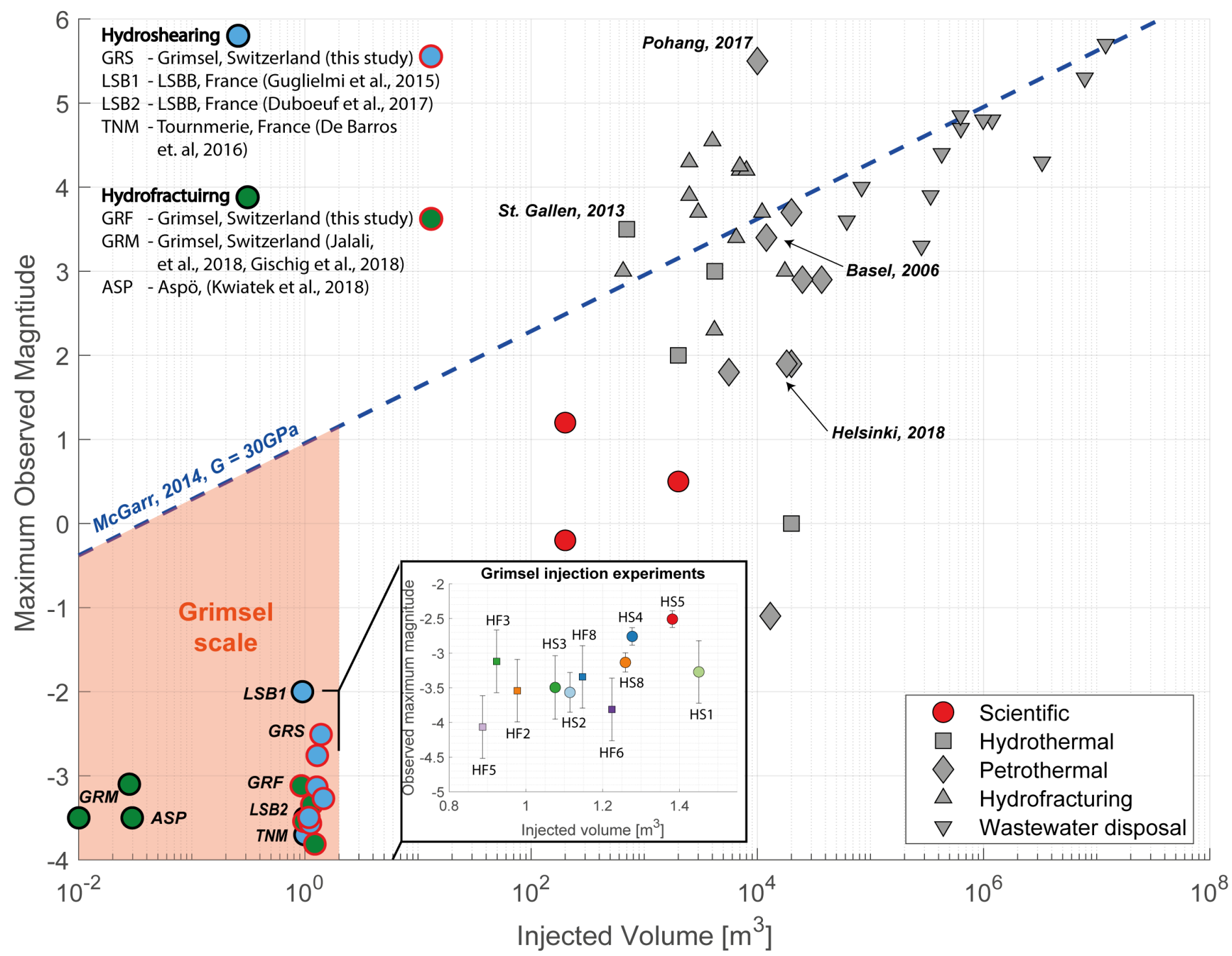

**Figure 1: Injected fluid volume vs. maximum observed magnitude of fluid injections at different scales, along with McGarr's (2014) estimate of the maximum observed seismic magnitude with respect to the injected volume. The detail box shows the maximum observed seismic magnitudes induced by the Grimsel injection experiments with respect to injected volume (error bars represent the standard deviation of all magnitude estimates of the respective seismic event). The magnitudes and injected volumes of larger scale injections (>100 m$^3$) directed towards hydrothermal, scientific and petrothermal purposes are adopted from Evans et al. (2012), injections directed towards waste water disposal are adapted from McGarr (2014), the projects directed towards hydrofracturing are adopted from Atkinson et al. (2016). Magnitude and injected volume data are from the hydrothermal project in St. Gallen (Obermann et al., 2015). Magnitudes and injected volumes for the petrothermal projects in Basel, Pohang and Helsinki are from Häring et al. (2008), Grigoli et al. (2018) and Kwiatek et al. (2019), respectively.**

*Scaled experiments for understanding stimulation processes*

To advance our understanding of behaviors related to reservoir stimulation and induced seismicity, scaled experiments have been conducted in underground laboratories worldwide. Smaller scales (~10 to 100 m) offer increased accessibility, controllability and repeatability compared to larger activities at field scales (~1000 m). Despite limitations due to the differences in stress field, temperature, etc., decametre- to hectometer-scale underground laboratory experiments are seen as a way to bridge the decametre-scale in the laboratory and the field scale.

Two HF experimental series in crystalline rock were recently performed in relation to EGS development. The Collab project (Schoenball et al., 2019) studied HF's in which notched locations in an injection borehole were used to control fracture initiation and the subsequent micro-seismicity was recorded by a surrounding dense seismic monitoring network. The experiments were performed at the Sanford Underground Research facility (SURF,

(Kneafsey et al., 2018)), which sustains an overburden of ~1500 m. Preliminary seismicity results show several strands of activated fracture plains mainly oriented perpendicular to the minimum principal stress. Another series of experiments was conducted at the Äspö Hard Rock Laboratory (overburden: ~410 m) in Sweden, to study the initiation of HF's using various water injection schemes. These schemes were designed to produce lower radiated seismic energy, while simultaneously optimizing fracture initiation and growth during the stimulation process (Zang et al., 2016). In addition to a reduction in the fracture breakdown pressures, they also observed a systematic reduction of $M_{max_obs}$ when using the advanced injection schemes (Zang et al., 2018). Kwiatek et al. (2018) document the spatiotemporal migration of seismicity towards and away from the injection interval. Radiated seismic energy correlates well with the hydraulic energy rate, but the total radiated seismic energy was low compared to the hydraulic injection energy. Fault plane solutions for particularly well-characterized seismic events mainly suggest that pre-existing fractures are reactivated during HF propagation. The experiments at Äspö were performed in a rock mass with strong geological variations (i.e., varying composition between granite, syenite, diorite and gabbro) in ductile and complex brittle structures (Zang et al., 2016). Experiments with similar injection volumes compared to the Äspö HF experiments were conducted at the Grimsel Test Site, as part of the stress measurement campaign performed prior to the experiments presented in this study (Krietsch et al., 2018b;Gischig et al., 2018).

Several HS experiments were conducted, but not in crystalline rock; during a fault reactivation experiment in limestone, fluid was injected into a pre-existing fault located at the Low Noise Underground Laboratory (LSBB) in France (overburden of 280 m). A step-rate injection method for fracture in situ properties (SIMFIP) instrument was installed across the fault zone; in addition to recording the seismicity and injection parameters (i.e., injection rate, pressure) it enabled distinguishing between the initial aseismic and subsequent seismic deformation phases during fault reactivation. Hydro-mechanical modeling further showed that about 70% of the permeability increase can be assigned to the initial aseismic phase (Guglielmi et al., 2015). Additional HS experiments performed on a normal fault at LSBB suggest that injection-induced aseismic motion plays a crucial role for pressure diffusion and the distribution of seismicity (Duboeuf et al., 2017). Experiments conducted in different areas of a shale fault zone at the underground research laboratory of Tournemire, France at an overburden of 270 m, induced deformation that was manly aseismic and the seismicity occurred mostly on calcified structures, leading to the conclusion that the number of induced seismic events might depend on the density of such fractures (De Barros et al., 2016). Taking the above mentioned experimental series in limestone and shale materials into account, De Barros et al. (2019) suggest that for an accurate prediction of seismic energy release during hydraulic stimulation, residual strain measured at the injection interval should be taken into account. Finally, an *in situ* fault reactivation experiment was performed in a clay-rich fault zone at the Mont Terri Laboratory, Switzerland, to assess the natural sealing integrity of CO2 sequestration sites (Guglielmi et al., 2017). Further investigation on the performed fault slip experiment in Mont Terri provided a detailed investigation on the rotation of slip vectors during fluid injection (Kakurina et al., 2019).

*Study objectives*

In this paper, we present observations of induced seismicity during twelve hydraulic stimulation experiments in a decameter-sized volume in crystalline rock at the Grimsel Test Site, GTS (Amann et al., 2018). We focus on investigating the influence of the local geological conditions, in connection with the prevailing stress field, on the

seismic response to high-pressure fluid injection. Six injection experiments were aimed at inducing shear dislocation on pre-existing structures (i.e., the hydro-shearing, HS- experiments) and six injection experiments were aimed at initiating and propagating tensile fractures in the intact rock mass (i.e., the hydro-fracturing, HF-experiments). To maintain consistency between the stimulation experiments, standardized injection protocols were used for both the HS and the HF experiments (Amann et al., 2018). Our monitoring and characterization program included building a geological model (Krietsch et al., 2018a) as well as detailed stress measurements (Krietsch et al., 2018b;Gischig et al., 2018;Jalali et al., 2018). Detailed rock mass characterization as well as high-resolution monitoring of the seismic, hydrological and mechanical response during stimulation allowed us to observe processes at high spatial and temporal resolutions.

The specific objectives of our work were to:

1) Document the seismological aspects of the GTS stimulation experiments and derive a high-quality catalog of earthquakes induced during the experiments.
2) Analyse the space-time evolution of seismicity and its relationship to injection parameters.
3) Interpret the induced seismicity, combined with other multi-parameter observations (strain, pressures, velocity changes) and geological characterisations with the aim to better understand and model the relevant mechanical and hydraulic process interactions during reservoir stimulation.
4) Study implications for safe and sustainable EGS reservoir development as well as managing the induced seismicity.

Our contribution focusses on the seismic response, which is linked to the hydro-mechanical observations during the six HS experiments by Krietsch et al. (in preparation-a), the six HF experiments (Dutler et al., 2019) and to study the permanent changes in the reservoir's hydraulic behavior (Brixel et al., under review).

## 2 The study site

The In-situ Stimulation and Circulation (ISC) project was carried out at the Grimsel Test Site (GTS), Switzerland. The underground research facility is operated by Nagra (i.e., the National Cooperative for the Disposal of Radioactive Waste). The test volume in the south of GTS has an overburden of ~480 m. It is intersected by two major shear zone types that are accessed by 12 boreholes for measuring the seismic, hydraulic and mechanical response to high pressure fluid injections (Figure 2; note: only the injection boreholes (INJ1, INJ2), one strain monitoring borehole (FBS2) and the stress measurement borehole (SBH4) used in this study are shown). In the following, the main features of the geological settings, the in-situ stress state and the experimental setup are summarized. For more details on the in-situ stress state, see Krietsch et al. (2018a), for the geological dataset and model see Krietsch et al. (2018b) and for the experimental setup refer to Doetsch et al. (2018a).

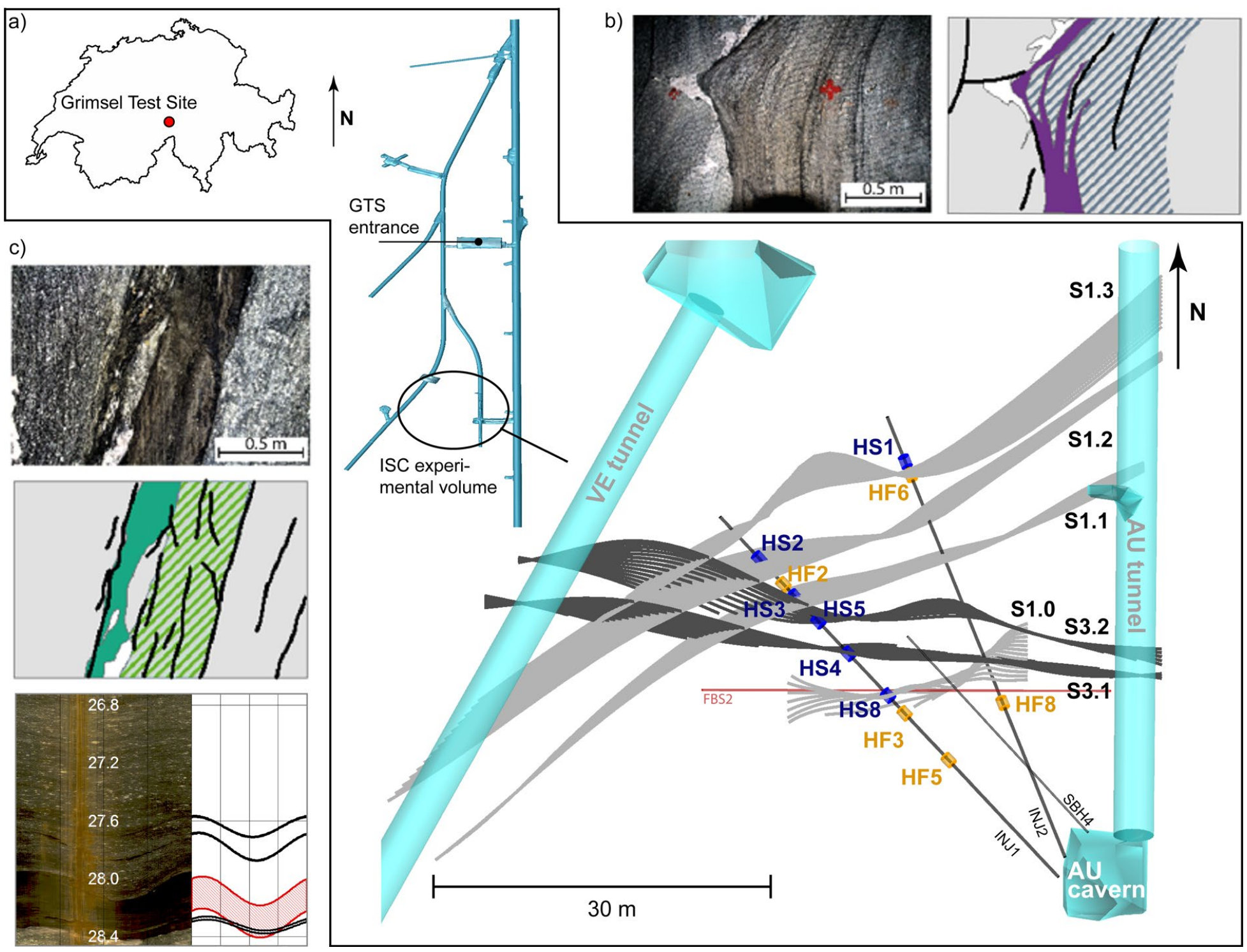

**Figure 2: a) Location of GTS in Switzerland (source: www.d-maps.com) and the location of the ISC experimental volume in the tunnel network operated by NAGRA, along with the top view of the ISC experimental volume located between the AU and VE tunnel. The two major shear zones S1 (grey) and S3 (black) intersect the experimental volume and the two injection boreholes (INJ1, INJ2) drilled from the AU-cavern. The location of the HS (blue) and HF (orange) injection intervals are shown, as well as the strain monitoring borehole FBS2 and stress measurement borehole SBH4 used in this study. b) Shear zone S1 observed in the AU-tunnel. (c) Shear zone S3 observed in the AU- tunnel along with its observation in the injection interval (red, adjacent fractures in black) of experiment HS4. (b, c) were modified after Krietsch et al. (2018a).**

The GTS is located within the Central Aare massif, at the lithological boundary between Central Aare Granite and Grimsel Granodiorite. The rock mass in the test volume has a relatively low fracture density and a foliation with

an average orientation of 140°/80° (dip direction/dip). Within the test volume, four shear zones with a ductile deformation history (referred to as S1with an orientation of 142°/77°) are characterized by a more distinct foliation compared to the host rock. These shear zones are associated with a few brittle fractures of various orientations that formed during retrograde deformation. In addition, two shear zones with a brittle-ductile deformation history (re-ferred to as S3, 183°/65°) are associated with biotite-rich metabasic dykes up to 1 m thick. The lateral distance between the two S3 shear zones is about 2.5 m and the rock mass between the faults is heavily fractured with more than 20 fractures per meter in the eastern section of the test volume. The different shear zones were labelled with an increasing index number, counted from South to North (i.e. S3.1 is south of S3.2, which belong to the S3 group, Krietsch et al. (2018a).

The stress characterization revealed a possibly unperturbed stress state about 30 m south of the S3 shear zone with principle stress magnitudes of $\sigma_1$ = 13.1 MPa, $\sigma_2$ = 9.2 MPa, $\sigma_3$ = 8.7 MPa and dip direction/dip of 104°/39° ($\sigma_1$), 259°/48° ($\sigma_2$) and 4°/13° ($\sigma_3$). The stress state close to the S3 shear zone is perturbed in terms of the principal stress magnitude and orientations. The minimum principal stress decreases to 2.8 MPa and the maximum principal stress direction rotates to 134/14° as the S3 shear zones are approached (Krietsch et al., 2018b). An overview of the mechanical material properties of the different species of granite found at the GTS is given in Selvadurai et al. (2019).

## 3 Methods

Six HS experiments were performed in February 2017 and six HF experiments were carried out in May 2017. Table 1 summarizes the details of each fluid injection in a chronological manner. The 12 injection intervals were chosen based on optical televiewer images taken in the two injection boreholes (INJ1, INJ2, Figure 2) and the geological 3D model introduced by Krietsch et al. (2018b). For the hydraulic shearing experiments, four of the chosen intervals targeted S1 structures (Figure 2, HS1, HS2, HS3, HS8). Two injections were performed on S3 structures (HS4, HS5). The injection intervals had a length of one or two meters and covered the target structure and adjacent brittle fractures (see example OPTV logs in Figure 2b, c). The hydraulic fracturing experiments were performed in intervals without observable fractures. Three experiments were performed to the south of S3 (Figure 2a, HF3, HF5, HF8) and two experiments were performed north of S3 (HF1, HF2). The exception is the HF6 experiment, which was planned to be performed in a fracture-free interval, but was conducted erroneously in a 1 m interval that contained S1.3 structures. Thus, the S1.3 structure stimulated during experiment HS1 was possibly re-stimulated during experiment HF6. Furthermore, during the initial injection experiment HF1, faulty shielding of a power line connecting the frequency control with the electric motor of the pump led to increased electronic interference on the seismic recordings and made further analysis impossible.

Table 1: Overview hydraulic shearing and hydraulic fracturing experiments

| **Experiment** | **HS2** | **HS4** | **HS5** | **HS3** | **HS8** | **HS1** | **Experiment** | **HF1** | **HF3** | **HF2** | **HF5** | **HF6** | **HF8** |
|---|---|---|---|---|---|---|---|---|---|---|---|---|---|
| **Date** | 08.02.2017 | 09.02.2017 | 10.02.2017 | 13.02.2017 | 14.02.2017 | 15.02.2017 | **Date** | 15/16.05.2017 | 16.05.2017 | 17.05.2017 | 17.05.2017 | 18.05.2017 | 18.05.2017 |
| **Target shear zone** | S1.2 | S3.1 | S3.2 | S1.1 | S1.0 | S1.3 | **Location with respect to S3** | north | north | south | south | No HF, targeted S1.3 | north |
| **Brittle fractures in interval** | 5 | >3 | >1 | 2 | 2 | 3 | **Brittle fractures in interval** | 0 | 0 | 0 | 0 | 2 | 0 |
| **Interval length [m]** | 2 | 1 | 1 | 1 | 1 | 1 | **Interval length [m]** | 1 | 1 | 1 | 1 | 1 | 1 |
| **Depth along borehole [m]** | 38.0 - 40.0 | 27.2 - 28.2 | 31.2 - 32.2 | 34.3 - 35.3 | 22.0 - 23.0 | 39.8 - 40.8 | **Depth along borehole [m]** | 40 - 41 | 19.8 - 20.8 | 35.8 - 36.8 | 14.0 - 15.0 | 38.4 - 39.4 | 15.2 - 16.2 |
| **Volume injected [m$^3$]** | 1.115 | 1.277 | 1.382 | 1.076 | 1.259 | 1.450 | **Volume injected [m$^3$]** | 1.156 | 0.924 | 0.978 | 0.887 | 1.224 | 1.147 |
| **Detected seismic events** | 1202 | 5607 | 2452 | 303 | 3703 | 559 | **Detected seismic events** | NaN | 1997 | 2204 | 1969 | 92 | 722 |
| **Located seismic events** | 63 | 3103 | 632 | 53 | 450 | 56 | **Located seismic events** | NaN | 70 | 519 | 13 | 15 | 183 |
| **Max. observed magnitude $M_A$** | -3.57 | -2.76 | -2.51 | -3.50 | -3.13 | -3.27 | **Max. observed magnitude $M_A$** | NaN | -3.12 | -3.54 | -4.07 | -3.81 | -3.34 |
| **b-value** | 1.69 ±0.26 | 1.36 ±0.04 | 1.03 ±0.05 | 1.93 ±0.37 | 1.61 ±0.12 | 1.93 ±0.39 | **b-value** | NaN | 1.55 ±0.26 | 1.35 ±0.08 | NaN | NaN | 2.66 ±0.36 |
| **Seismogenic index** | -5.8 | -3.0 | -2.4 | -7.6 | -4.9 | -6.6 | **Seismogenic index** | NaN | -4.8 | -4.0 | NaN | NaN | -9.0 |
| **Seismically activated area [m$^2$]**[1] | 68.5 | 210.8 | 284.7 | 97.9 | 112.8 | 137.4 | **Seismically activated area [m$^2$]**[1] | NaN | NaN | 94.6 | 8.0 | NaN | 235.7 |
| **Injection efficiency**[2] | 3.99e-05 | 2.45e-04 | 6.29e-05 | 7.46e-05 | 1.03e-04 | 5.81e-04 | **Injection efficiency**[2] | NaN | NaN | 2.79e-05 | NaN | NaN | 3.08 |
| **Ratio seismic deformation**[3] | 1.5e-3 | 7.7e-3 | NaN | 1.3e-3 | 3.7e-3 | 1.8e-2 | **Ratio seismic deformation**[3] | NaN | NaN | NaN | NaN | NaN | NaN |

[1]Mean area of convex and concave hull; [2,3] seismicity integrated to a magnitude of -9

## 3.1 Injection protocol

A standardized injection protocol was used for the six HS experiments, to compare the influence of the targeted geological structures on the seismo-hydro-mechanical response. Roughly 1 $m^3$ of fluid volume was injected per experiment (actual volumes are given in Table 1). The injection protocol consisted of four injection cycles (referred to as C1 – C4, Figure 4), in which either the injection pressure or injection flow rate was increased in a stepwise manner after steady-state was reached. All the cycles were followed by a shut-in phase, where pumping was stopped and a venting phase, in which the pressure in the injection- and all monitoring-intervals were bled off. The first two pressure-controlled cycles C1 and C2 were conducted to determine pre-stimulation jacking pressure (i.e., the injection pressure at which the ratio between the injection pressure and flow rate deviates from constant) and initial injectivity of the target structure. C3 was the actual flow-controlled stimulation cycle, in which the bulk part of the fluid was injected. C4 was initially pressure controlled, changing to flow controlled injection, aimed at determining the post-stimulation jacking pressure and injectivity of the targeted structure. During all HS experiments, the flow rate did not exceed 38 l/min.

HF experiments also followed a standardized injection protocol involving a target injected volume of ~1 $m^3$ (actual injected volumes in Table 1). The injection protocol for the five HF experiments started with a flow-controlled formation breakdown cycle (indicated by the letter F) to initiate the hydraulic fracture. This initial cycle, and all the subsequent cycles, included a shut-in phase were pumping was stopped. During some of the formation breakdown cycles the shut-in phase was complemented by a bleed-off phase of the injection interval and all pressure monitoring intervals. The two subsequent cycles were aimed at propagating the previously initiated hydraulic fracture (RF1, RF2). For these two propagation cycles, water was used during HF1, HF2, HF3 and, for HF5, HF6 and HF8, shear thinning fluid (xanthan-salt-water mixture, XSW) was used. We note that the XSW mixture exhibited a viscosity of ~35 cPs (viscosity of water = 1cPs). Propagation cycles RF1 were performed with maximum flow rates of 35 l/min. During experiments performed with water, the flow rate was controlled in a sinusoidal fashion (period: 2.5 – 20 s, amplitude: +/- 15 l/min) for roughly 10 minutes. For experiments in which XSW was injected, an additional cycle, RF3, was added by injecting fresh water, in some experiments with cyclic flow rates, allowing flushing out the XSW. All the HF experiments were finalized by a pressure-controlled step-rate injection test (SR) for evaluating post-stimulation jacking pressures and injectivities of the created hydraulic fracture.

## 3.2 Seismic monitoring and data processing

### 3.2.1 Seismic monitoring

A total of 26 in-situ acoustic emission sensors (AE sensors) formed the passive seismic network around and inside the test volume (Figure 3a, green cones). The sensors were manufactured by the Gesellschaft für Materialprüfung und Geophysik GmbH (GMuG), and have a bandwidth of 1 to 100 kHz, with their highest sensitivity at 70 kHz. 14 AE sensors were installed on a tunnel level (R2 - R15, type: Ma-Bls-7-70), in 55 mm diameter boreholes drilled approximately 250 mm deep into the tunnel wall. The bottom view AE sensors were pressed against the polished surface of the base of the borehole. The core of the network (i.e. sensors within 5-25 m distance to the injection intervals) was composed of eight borehole AE sensors (R16 – R23, type: MA-BLw-7-70-68) distributed in four water-filled monitoring boreholes (GEO1- GEO4, Figure 3a). The borehole AE sensors have a curved front surface and were deployed in sensor-shuttles in which two pneumatic cylinders (line pressure 10 bar) ensured contact pressure between the sensors and the borehole wall. Four additional sensors (R24 – R27, type: MA-BLw-7-70-86)

with curved front surfaces were installed in borehole SBH4 (Figure 2b). For calibration purposes, five one component (1C) accelerometers (R28 – R32, type: Wilcoxon 736T) were installed next to five of the tunnel level AE sensors (Figure 3a, red cones, R4, R6, R7, R9, R11). The accelerometers were factory calibrated and feature a flat frequency response from 50 to 25’000 Hz, with a sensitivity of 100 mV/g. They were mounted to brass disks (Ø28 x 1 mm), which were glued to the front surface of the 55 mm diameter and 100 mm deep boreholes drilled into the tunnel wall adjacent to the AE sensors (Figure 3c). The seismic signals were recorded continuously on a 32 channel acquisition system at a 200 kHz sampling rate (GMuG, digitizer cards: Spectrum M2i.47xx). AE sensor channels had 1 kHz and accelerometer channels had 50 Hz high-pass analogue filters installed.

In addition to the passive seismic network, active seismic sources were installed; eight falling hammer sources were distributed in the AU- and the VE-tunnels. Two borehole piezoelectric sources were installed in borehole GEO2 and GEO4 (Figure 3a, black arrows). The trigger signal of the seismic sources, used to determine the initiation time of each active seismic survey, was recorded on one channel of the acquisition system. The active sources were used for time-lapse 3D seismic tomography surveys during the experiments (Schopper et al. (under review);Doetsch et al. (2018b) for details). For more information on the seismic monitoring system, see Doetsch et al. (2018a) .

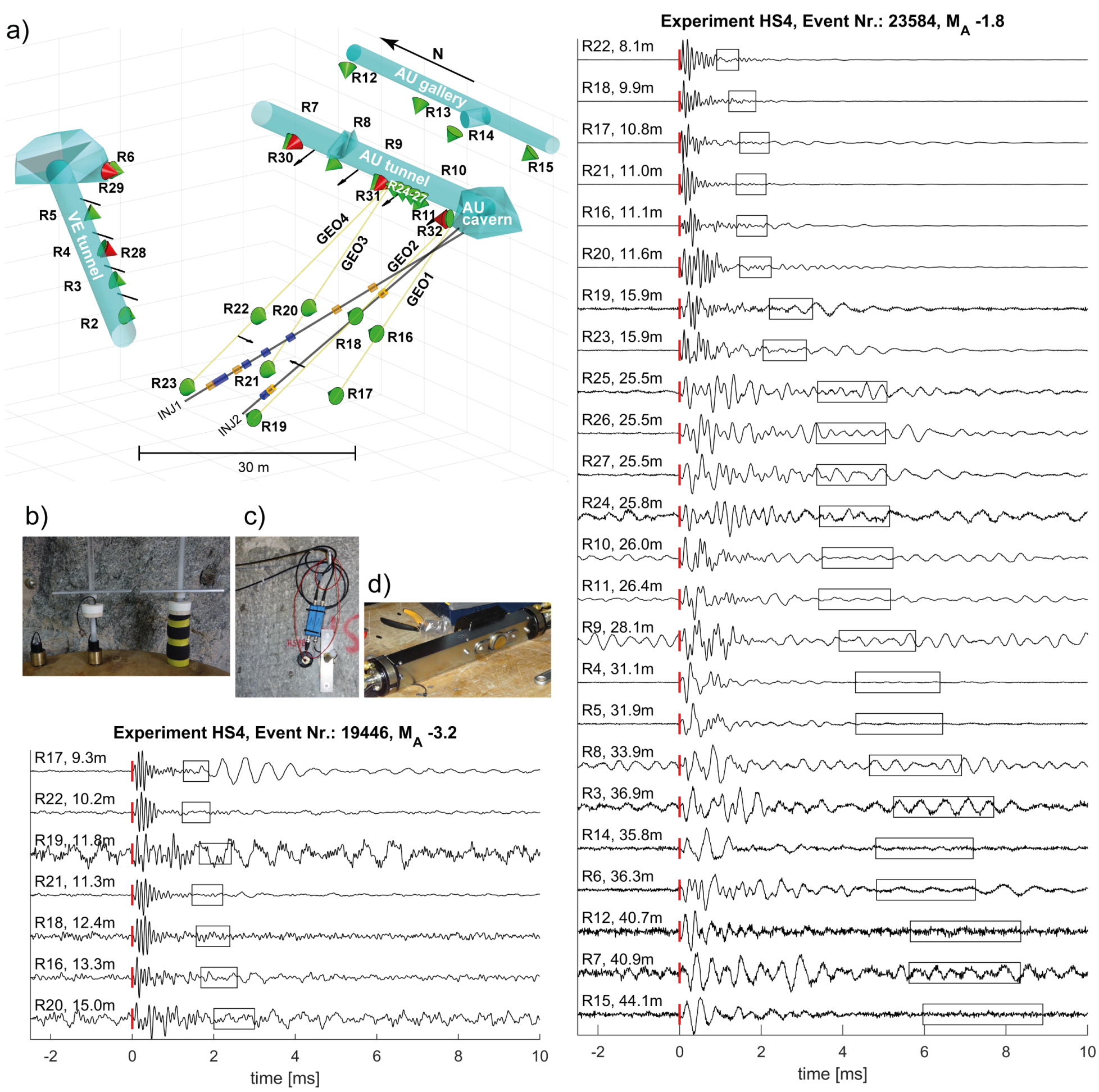

**Figure 3: a) The seismic network consisting of 26 uncalibrated AE sensors (green cones) installed in four boreholes, the AU- and VE-tunnels as well as the AU-gallery, along with five one component calibrated accelerometers (red cones) collocated with five AE sensors in the AU- and VE-tunnels. Seismic sources in the tunnels and boreholes are shown by black arrows. b) AE tunnel sensor, insulated against acoustic noise. c) Installed AE sensor next to a calibrated accelerometer along with their pre-amplifiers. d) AE sensors in a sensor-shuttle for deployment into the formation's water filled borehole. Waveforms from a small and large magnitude event induced during experiment HS4, including the Euclidean distance hypocenter - sensor, P-wave pick (red-stripe), and a window of a hypothetical S-wave arrival for an S-wave velocity of 2500 - 3000 m/s (applied bandpass filter for small event: 1-12kHz, large event: 1-50kHz).**

### 3.2.2 Seismic data processing

Continuous recording of 32 channels at a sampling rate of 200 kHz with 16-bit digital resolution resulted in ~250 GB of data over approximately 6 hours of recording time. For flexible and fast access to the data, the Adaptive Seismic Data Format (ASDF, Krischer et al. (2016)) proved to be adequate. The ASDF format is integrated in an open source Python library for seismology (ObsPy) that was used for event detection.

For seismic event detection only the eight closest AE sensors to the center point of the injection interval were considered, (i.e., R16 – R23). Prior to any event detection, the data streams were bandpass filtered (4th order Butterworth filter) between 1 kHz and 12 kHz. An ObsPy integrated detection algorithm with a recursive STA/LTA trigger and a coincidence threshold of 2 was used for event detection (i.e. a seismic event was declared when at least two detections of a potential seismic event were found). Many of the triggered events were electric noise interference characterized by their high frequency and near-simultaneous occurrence on all channels. These events were automatically removed, if the trigger time of the recursive STA/LTA algorithm or the time of the minimum-, or the maximum-amplitude was within 4 sample points. The event catalogues produced with sensitive trigger settings were inspected visually to remove false events (e.g. electric noise, man-made signals produced in the tunnels, etc.). Note that throughout the experiments, active seismic surveys were performed approximately every 10 minutes. During the perturbance by the active seismic signals (i.e., 1 s for each hammer source, 35 s per piezo-electric source (TRBLw-1-86) burst) no passive event detection was performed (see also a detailed temporal evolution of seismic event detections, initiated active seismic signals and injection parameters of all the experiments in the supplementary material SM1).

P-wave onsets were manually picked for events with coincidence levels three to eight (i.e. the signal was detected on three to eight traces). As can be seen from the seismic events detailed in Figure 3, S-wave signals were generally weak or undistinguishable (a possible reasoning can be found in section 3.7). Thus, the S-wave onsets could not be picked and used for event location.

### 3.2.3 Seismic event location

The seismic events were located using a homogeneous, transversely isotropic velocity model and standard inversion practice. The P-wave arrival times were weighted according to their P-wave pick uncertainties, which were estimated as a function of signal to noise ratios (SNR). The SNR was calculated from the maximum absolute P-wave amplitudes determined in a window defined by the P-wave onset and a theoretical S-wave onset (estimated with an S-wave velocity of 2800 m/s), as well as the maximum absolute amplitude in a noise window taken in a window with the same length before the P-wave onset. At an SNR > 30, P-wave pick uncertainties were estimated at plus/minus two samples, below a ratio of 30 P-wave pick uncertainties (in samples) were estimated with the following linear relationship:

$$\varepsilon_p = \pm 2 \; if \; SNR \geq 30$$
$$\varepsilon_p = -0.16\, SN + 8.8 \; if \; 30 > SNR \geq 5$$
$$\varepsilon_p = -2.5\, SN + 20.5 \; if \; 5 > SNR \geq 1 \qquad (1)$$
$$\varepsilon_p = -182\, SN + 200 \; if \; 1 > SNR$$

The anisotropic velocity model is based on the weak elastic anisotropy formulation of Thomsen (1986). Thomsen's formulation for transverse isotropy is:

$$v_P = v_{P,sym}(1 + \delta sin^2(\theta) cos^2(\theta) + \varepsilon sin^4(\theta)) \qquad (2)$$

where $v_p$ is the P-wave velocity along a respective ray path, $v_{P,sym}$ represents the P-wave velocity along the anisotropy symmetry axis (i.e., usually the minimum velocity), $\theta$ is the angle between the symmetry axis and the ray path, the parameter $\varepsilon$ represents the relative increase in velocity perpendicular to the symmetry axis and $\delta$ describes the angular dependency of the velocity. The best-fitting anisotropic velocity model (i.e. $v_{P,sym}$, $\varepsilon$, $\delta$, azimuth and dip of symmetry axis) was inferred with a Matlab genetic algorithm from a sub-catalog of 495 induced high-quality seismic events exhibiting more than nine P-wave picks and locations distributed over the entire experimental volume. For this, the median of the root-mean-square (RMS) of the differences between theoretical and observed arrival times for 495 high-quality event locations was minimized. Furthermore, to verify the estimated P-wave pick uncertainties the dimensionless chi for each of the 495 events in the sub-catalog was computed and did not exceed a value of 3.6.

$$chi = \sqrt{\frac{1}{N}\sum_{i=1}^{i=N}\left(\frac{d_i^{obs} - d_i^{pred}}{\varepsilon_p}\right)^2} \qquad (3)$$

Note that the target value for chi is 1.0, for which the discrepancy between the observed and predicted arrival times is equal to the estimated pick uncertainty. Values above 1.0 suggest an underfitting, values below 1.0 suggest an overfitting of the data.

Comparing the velocity parameter determined through the aforementioned analysis steps, with the seismic velocity parameter introduced by Gischig et al. (2018) at similar location at GTS, our inferred seismic velocity in the direction of symmetry, $v_{p,sym}$, is about 5.5 % lower, but the ratio between the two velocities, $\varepsilon$, remains the same. A slight change in the angular velocity dependency, $\delta$, was also observed (0.07 instead of 0.02). The dip direction and dip of the symmetry axis also changed slightly compared to Gischig et al. (2018) (310°/29° instead of 330°/20°). We attribute these differences to the geological conditions; the rock mass contained a highly fractured shear zone compared to the less fractured rock mass within the ISC test volume targeted by Gischig et al. (2018) for their mini-fracturing experiments. Station corrections were determined for each sensor location using the joint hypocenter determination approach analogous to Gischig et al (2018). These account for systematic shifts in travel times arising from error in sensor locations or geological conditions around the sensor.

To estimate location uncertainties of source locations due to pick uncertainties, the arrival times were randomly perturbed 1000 times with the estimated pick uncertainties (similar to Gischig et al. (2018)). The principal directions and dimensions of the point clouds consisting of the 1000 new locations were analysed to estimate the location relative errors. Only events with the largest error axis below 3 m (i.e., ± 1.5m) were analysed further.

Absolute location uncertainties were estimated by comparing the known initiation locations of high-energy sparker shots (i.e., high-voltage electric discharge, which triggers a compressional wave in formation water-filled boreholes) in injection boreholes and their inferred location through the determined velocity model and station corrections. The absolute location errors were below 0.5 m in injection borehole one (INJ1) in an interval from 15 to 30 m depth and increased to around 1.5 m towards the borehole top and bottom. For injection borehole two (INJ2) the absolute error was below 1 m in an interval from 15 to 30 m depth and increased to around 1.5 m towards the borehole mouth and bottom.

#### 3.2.4 Magnitude computation

Determining the magnitude of seismic events recorded on uncalibrated AE sensors is challenging. Angle dependent sensitivity variations and varying coupling quality make it impossible to infer a simple and universal instrument response (Kwiatek et al., 2011). However, to characterize the relative source strength of induced seismic events, relative magnitudes, $M_r$, were estimated from the maximum P-wave amplitudes of uncalibrated AE sensors in the time domain following the approach introduced by Eisenblätter and Spies (2000) in combination with an attempt to account for angle dependent sensitivity variations and variations in coupling quality. To adjust the estimated relative magnitudes $M_r$ to a realistic magnitude level, the absolute magnitudes $M_W$ are determined for events recorded on tunnel-level AE sensors collocated with calibrated accelerometers (Figure 3a, red cones). Adjusted relative magnitudes $M_r$ are referred to as adjusted amplitude magnitudes $M_A$. Relative magnitudes were estimated as follows:

$$M_r = log_{10}\left(\frac{1}{N}\sqrt{\sum_{i=1}^{N}\left(A_i\frac{r_i}{r_0}e^{a(r_i-r_0)}\right)^2}\right) \tag{4}$$

where $A_i$ is the band-pass filtered (3 – 12kHz) maximum P-wave amplitude determined in a window confined by the P-wave arrival pick and a theoretical S-wave arrival, assuming a shear wave velocity of 2800 m/s. $r_i$ is the source-sensor distance, $r_0$ is a reference distance (chosen to be 10 m) and $N$ is the number of P-wave arrivals of the respective event. $a = \pi f_0/(Q_P V_P)$ represents the frequency dependent attenuation coefficient, where $f_0$ is the dominant frequency, which was chosen to be the middle-frequency of the filtered band (i.e., 7.5 kHz), $V_p$ is the P-wave velocity and $Q_p$ is the quality factor representing seismic attenuation. $Q_p$ was measured at GTS by Holliger and Bühnemann (1996) in a frequency range of 50 – 1'500 Hz, and was reported as 20 – 62.5. More recently, Barbosa et al. (2019) estimated $Q_p$ from full waveform sonic data in the injection boreholes using sources in the range of 15 – 25 kHz. They found $Q_P = 13$ on average with a drop to the very low values of 8 in the vicinity of the metabasic dykes and the shear zones. Based on these observations, we chose a $Q_p$ value of 30. For the relative magnitude estimate, only tunnel sensors (R2-R15) and borehole sensors (R16-R23) were used.

##### a. Correction of angle-dependent sensitivity variation of AE sensors

The installed AE sensors at Grimsel are of similar type to the AE sensors used by Manthei et al. (2001), who observed a declining sensitivity with an increasing incidence angle of incoming seismic waves with respect to the sensor normal. The varying sensitivity is due to both the design of the sensor and the coupling quality of the sensor to the rock and thus cannot be dealt with in a generic manner, as is described by GMuG. The influence of the incident angle on the relative magnitudes of the incoming seismic waves has been characterized experimentally at the GTS using the two parallel boreholes GEO1 and GEO3 (Figure 3a). A piezoelectric source of the type TR-BLw-1-86 (manufactured by GMuG) was incorporated in the same shuttle as the AE sensors, radiating seismic

energy in a spectrum similar to the observed seismic events (1 to 15 kHz). The sensor was deployed in GEO3 at a fixed location in the direction of GEO1, while the source was placed in GEO1, and moved in 0.5 m increments, resulting in an incidence angle range from 0° to 50 °. The waveforms of 250 pulses per locations were stacked. From these signals, a relative magnitude $M_r$ was estimated revealing a linear decay of $M_r$ as the incident angle increased (see supplementary material SM2, a). Averaging the slope of 20 measurement series at 20 different locations along the boreholes GEO1 and GEO3, and accounting for any variation in coupling quality, leads to an angle-dependent $M_r$ correction function $Mr_{corr}(\alpha) = Mr + 0.0104 \cdot \alpha$, where $M_r$ is the relative magnitude estimated without correction and $\alpha$ is the incident angle of the direct incoming P-wave.

**b. Correction for variation in coupling quality of AE sensors**

To account for variations in the coupling quality of AE sensors during the actual stimulation experiments, a correction quantity was calculated for each AE sensor by iteratively minimizing the median of sensor residuals:

$$\Delta M_{r_i} = median(M_{r_{mean}} - M_{r_i}) \quad (5)$$

where $\Delta \mathrm{M}_{\mathrm{r}_i}$ is the median difference of the $i'th$-sensor, $M_{r_{mean}}$ is the mean relative magnitude of at least three sensors and $M_{r_i}$ is the relative magnitude estimate of the $i'th$-sensor (see supplementary material SM2, b).

After the application of the aforementioned corrections, standard deviations of the estimated $M_r$ are, for most of the seismic events, approximately 0.3 but can reach 0.7. Standard deviations are lower for events located in the focus of the seismic network (i.e., experiments HS4, HS5). Note that because we are lacking knowledge on the decline of sensitivity of AE sensors above a 50° incidence angle, $M_r$ was only estimated at AE sensors for which the incidence angles did not exceed 50°.

**c. Estimating instrument responses for AE sensors**

In order to establish the absolute magnitudes $M_w$ for a subset of located events, we determined the instrument responses for the five collocated AE sensor–accelerometer pairs installed on a tunnel level using the spectral deconvolution calibration technique introduced by Plenkers (2011) and Kwiatek et al. (2011). Based on their technique a calibration function, $Z(f)$, can be computed:

$$Z(f) = \frac{u^{AE}(f)}{u^{Acc}(f)} = \frac{i^{AE}(f)}{i^{Acc}(f)} \quad (6)$$

where, $u^{AE}(f)$ and $u^{Acc}(f)$ represent the displacement signals, $i^{AE}(f)$ and $i^{Acc}(f)$ are the instrument responses in the frequency domain of the acoustic emission sensors and the calibrated accelerometers, respectively. From the complex calibration function $Z(f)$, only the modulus of the relative amplitude calibration function $|Z(f)|$ is used. The calibration technique relies on seismic signals recorded on both the AE sensors and the collocated calibrated accelerometer. However, most of our induced seismic events were too weak to be recorded by the less sensitive accelerometer with adequate high-frequency quality. Therefore, instrument responses were inferred from the aforementioned high-energy sparker shots performed every 0.5 m in the boreholes INJ1, INJ2 and GEO1-4 (Figure 3a). Sparker shots radiate seismic energy in a similar frequency band as the induced seismic events (~1-50 kHz).

To infer instrument responses, four milliseconds of the waveform centered around the first P-wave arrival from performed sparker shots were used (excluding clipped signals and signals with an SNR ratio smaller than 10 dB). Before computing the Fourier spectra, the waveforms were bandpass-filtered (AE sensors: 1 – 50 kHz, accelerometer: 1 – 25 kHz), zero padded and tapered with a Hanning window. Signal and noise spectra were smoothened

using a Savitzky-Golay filter (polynomial order: 3, frame length: 51). The maximum frequency considered for the instrument response is the one that still had a signal 3 dB above the noise floor.

Instrument responses were calculated for 10 sparker shots per incidence angle bins of 15° up to incidence angles of 60° , since it was suggested by Kwiatek et al. (2011), Plenkers (2011) and Naoi et al. (2014) that the instrument responses are incidence angel dependent. However, no angle dependency could be resolved for our sensor pairs, perhaps because both the AE sensors and 1D-accelerometer were oriented in the same direction and the angle dependent sensitivity variations cancelled out. We note that, compared to the studies that showed sensitivity variations with changing incidence angles, the incidence angle of seismic events in our study (i.e., sparker shots in our case) differed in spatial scale. In this research, we were limited to a rather narrow band and did not exceed 60° because the AE sensor - accelerometer pairs installed at the tunnel level were aligned towards the injection intervals (see the geometric details shown in Figure 3a). Since we did not observe angle dependent variations in the instrument responses, we used the ten instrument responses that exhibited the largest frequency range, and found no difference in the incident angle of the direct P-wave. In contrast to the incidence angle dependency of instrument responses, distinct variations in instrument responses for the different collocated AE sensor - accelerometer pairs were observed (see supplementary material SM2, c), which is possibly due to different coupling qualities of the sensors. Thus, it is impossible to transfer instrument responses for other AE sensors installed at the tunnel level to those down-borehole. We have therefore only calculated the absolute magnitudes $M_W$ as determined for the AE sensors (R4, R6, R7, R9, R11) collocated by the accelerometers (R28-R32).

### d. Estimating absolute magnitudes $M_W$ for a subset of events

For corrected P-wave source spectra recorded on AE sensors R4, R6, R7, R9 and R11 exhibiting a SNR > 10 dB moment magnitudes were determined by fitting the theoretical displacement source spectrum introduced by Boatwright (1978), corrected for aseismic attenuation and geometrical spreading to the observed spectra:

$$\Omega_P(f) = \frac{\Omega_{0,P}}{1+\left(\frac{f}{f_c}\right)^2}\ exp\left(-\frac{\pi R f}{Q_P v_P}\right)\frac{1}{R} \tag{7}$$

where, $\Omega_{0,P}$, is the low frequency plateau of the P-wave spectrum, $f_c$, represents the corner frequency. where, $Q_P$, is the frequency-independent quality factor (again set to 30) and, $v_P$, represents the P-wave velocity (chosen to be 5030 m/s, mean anisotropic velocity), $R$, is the source – sensor distance. The scalar seismic moment is then derived from the low frequency plateau using:

$$M_0 = \frac{4\pi\varrho v_P^3 \Omega_{0,P}}{R_P \Upsilon_P} \tag{8}$$

Here, $\varrho$, represents the density of the rock mass and is chosen to be $2650\ \frac{kg}{m^3}$, the radiation pattern correction factor, $R_P$, is set to 0.52 and the free surface correction factor, $\Upsilon_P$, is chosen to be 2 (Aki & Richards, 2002). The scalar seismic moment is converted into a moment magnitude using the relation $M_W = \frac{2}{3} log_{10}(M_0) - 6.03$. The theoretical spectrum $\Omega_P$ was fitted to the observed spectrum using a grid-search varying $M_0$ and $f_c$, keeping $Q_P$ constant. $M_W$ were estimated for events with at least two $M_W$ estimates. Comparing the obtained $M_W$ with $M_r$ leads to the relationship for the amplitude magnitude $M_A = M_r - 4.0$.

# 4 Results

In the following, we present and compare seismicity observed during the ISC stimulation experiments. Seismicity is in two cases combined with i) strain observations and ii) inferred velocity changes in the experimental volume, in order to show the diversity and interaction of the observed properties.

## 4.1 Temporal seismic event evolution

For the HS injection experiments, most events (12'211 from a total 13'826 detections) were detected during pumping phases (Figure 4a, b). The percentage of events recorded during shut-in were in the range of 10%. Less than 2% of events were detected during venting. Comparing the HS experiments, significantly fewer events were detected during experiments at the S1 shear zones compared to the stimulation experiments performed in the S3 shear zones (Table 1). An exception was HS8 at the S1 shear zone south of S3, which produced a number of events comparable to the S3 injections (i.e., total detections: 3703). This may be explained by the fact that the injected fluid entered the S3 shear zone, which was evident from the seismicity cloud migrating towards the S3 shear zone (see section 3.2). The number of seismic events (normalized to the total number of events per experiment) is plotted against injected volume in Figure 5a, b. Again, a distinct behavioural difference between S1 and S3 injections is observed. During experiments in S1, the largest seismic detection rate was observed during stimulation cycle 1; more than 50% of all events were induced with less than 100 liters of fluid (<10% of the total volume). On the contrary, for S3 stimulations, most events were detected during cycle 3, when the largest volume of fluid was injected. Again, experiment HS8 is an exception in that the highest detection rate was observed during cycle 1 (similar to S1 stimulations), after which the event rates behave similarly to the S3 injections (HS4, HS5). Generally, a larger fraction of seismic events occurred after shut-in during injection into the S1 shear zones compared to injections into the S3 shear zones.

During the HF injections, significantly fewer detections were made compared to the HS injections (Figure 4c, d). Most of the events were detected during the pumping phases (4'483 of 6'731 detections). A comparably high percentage of detections (33%) were made during shut-in and no events were detected in the venting phases. Possibly, the high percentage of post-shut-in detections were related to the hydraulic connection created between the injection interval and the open seismic monitoring boreholes (termed GEO) during the last two experiments HF5 and HF8. We observed that the hydraulic connection allowed flow through GEO boreholes towards the tunnel, and assume that this triggers stick-slip movements of the AE sensors. Thus, ongoing flow through GEO boreholes after shut-in would explain the large number of post-shut in events (see also detection rate per volume in Figure 5b). Note that HF6 - by mistake placed across the S1.3 shear zone close to the injection interval of HS1 - can be seen as a continuation to the HS1 experiment.

In summary, for the HS experiments, 31% (i.e., 4342) of detected events could be located. The fraction of located shut-in events during the HS experiments is around 3%, the fraction of events induced during the venting phase is less than 1%. For the HF experiments, because of the large number of events without seismic origin (possibly sensor stick-slip), only 12% (i.e., 781) of all detected events could be located. 6% of the events were located after shut-in and no events were located during the venting phases. The located seismic events fulfill a location uncertainty below ± 1.5m (for more information on location uncertainty see section 2.2.3).

The maximum induced magnitudes $M_{max}$ during both HS and HF experiments (see inset of Figure 1 and yellow stars in Figure 4 and Figure 5) occurred during pumping with no evidence of a temporal trend. Events during a time interval between shut-in and the start to a new injection cycle were usually of lower magnitude. One exception

was the injection experiment HF6; here the highest magnitude event was induced during a shut-in phase (see supplementary material SM3).

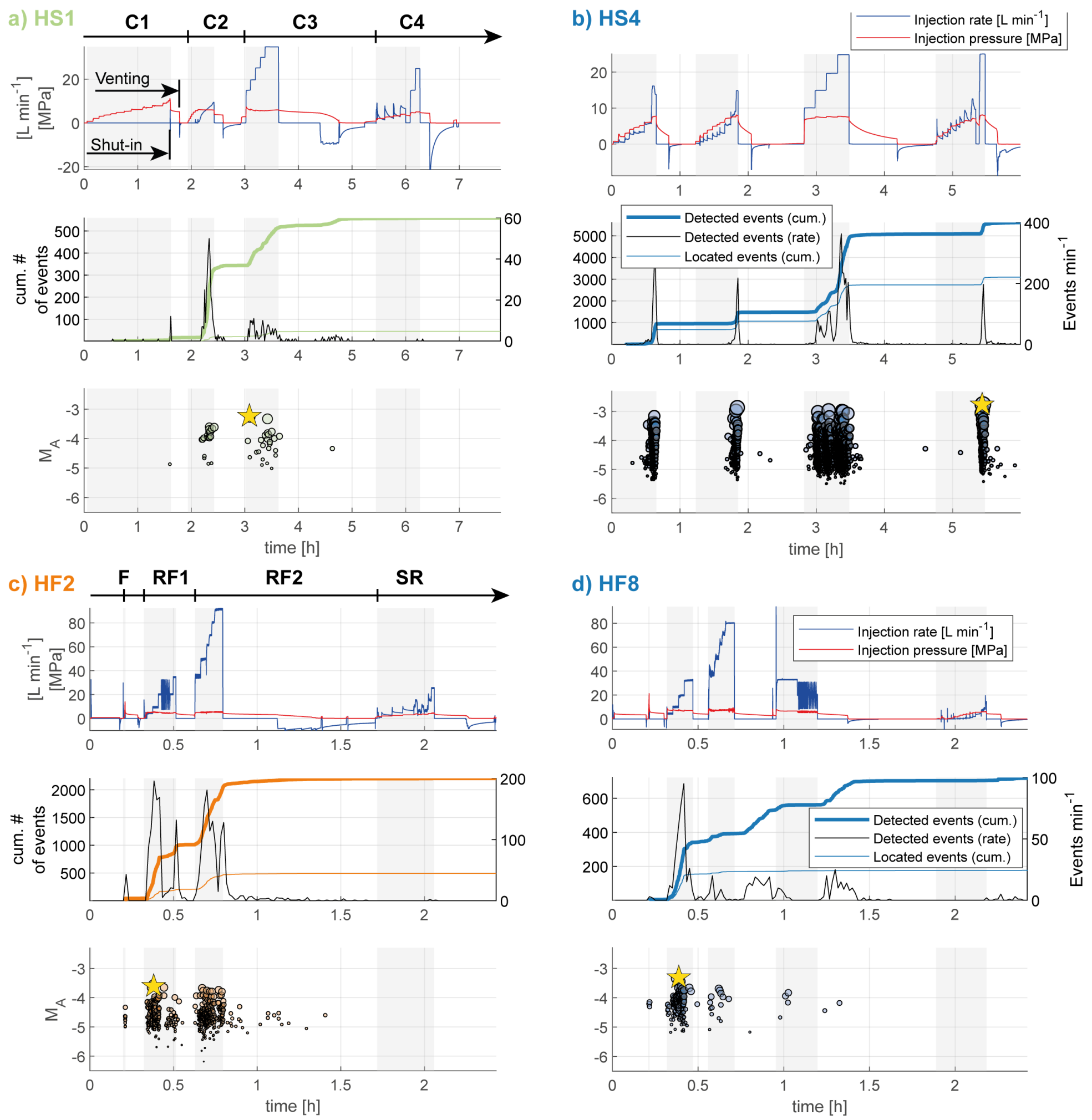

**Figure 4: a) Temporal event evolution of experiment HS1 performed in shear zone S1.3, b) of experiment HS4 in shear zone S3.1 along with c) the temporal event evolution of experiment HF2 which was performed north of the S3 shear zones and d) the temporal event evolution of experiment HF8 performed south of the S3 shear zones. In addition to the injection rate and pressure, the cumulative number of events and magnitudes $M_A$ are shown. The largest magnitude event is indicated with a yellow star. The shaded area on the plots indicate the pumping periods during an experiment (the temporal event evolution of the remaining experiments is shown in the supplementary material SM3 of this manuscript). a) also shows an example for a HS injection protocol with injection pressure and injection flow rate, divided into the four cycles, including shut-in and venting phases in each cycle. c) shows an example of a HF injection protocol with injection pressure and injection flow rate, including formation break down cycle (F), refrac cycles (RF) and the final step rate (SR) injection experiment. All of the cycles include a shut-in phase, but in some cycles only a venting phase is included.**

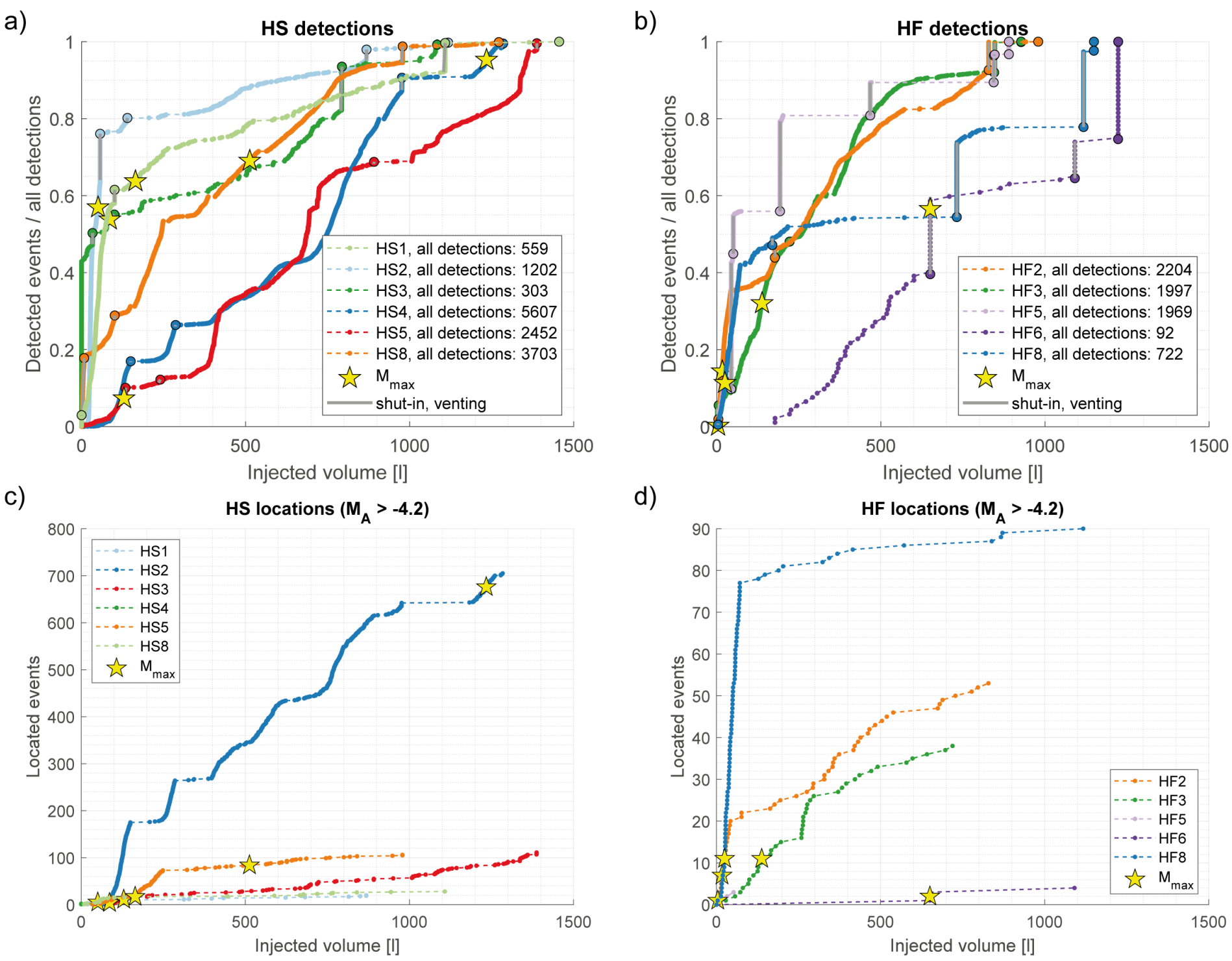

**Figure 5: Cumulative fraction of detected events as a function of cumulative injected volume of a) HS and b) HF injection experiments. c) and d) show absolute values of located events above a magnitude level of $M_A$ -4.2 for HS and HF experiments, respectively.**

## 4.2 Spatial properties of seismicity clouds

The seismicity clouds produced by the HS experiments (Figure 6a, c) form planes with a tendency to align in the EW direction (main direction of S3 shear zones) or in a NE - SW direction (main direction of S1 shear zones). Often these planes exhibit substructures with events grouped into clusters, which is most pronounced for experiment HS4 (see also Figure 7 and Figure 8). Note that we use the term "cluster" here for a distinct subgroup of seismic events within the seismicity cloud of individual experiments. These are not clusters derived from waveform similarity and relative relocation, which is the scope of future studies. The seismicity induced by the injection experiments in injection borehole INJ1 predominately propagated in an easterly direction, whereas the seismicity cloud of HS1, the only HS injection in INJ2, was oriented in a NE-SW direction (Figure 6a). For this experiment the seismicity occurred exclusively a few meters above the injection interval (Figure 6c). For HS8, the injection experiment closest to the top of injection borehole INJ1, there was a tendency for downward propagation. Generally, seismicity is well contained within narrow clouds surrounding the injection interval. However, interactions (i.e., hydraulic or mechanical) were evident in experiments HS4 and HS8, where part of the HS8 seismicity cloud aligns with the HS4 seismicity cloud.

The seismicity clouds of the HF injection experiments also had a tendency to propagate in the EW direction, similar to the HS experiments. Experiments conducted in INJ1 (i.e., HF2, 3, 5) induced seismicity clouds that propagated towards the East from the injection interval, whereas injection into INJ2 (i.e., experiment HF8) induced a seismicity cloud propagating towards the West (Figure 6b). HF6, the HF experiment misplaced at the S1.3 shear

zone, induced only a few seismic events superimposed on the seismicity cloud of experiment HS1 that targeted at the same structure. Seismicity clouds that occurred during the HF experiments propagated preferentially downwards. Injection experiment HF3 stands out in that it induced a dispersed seismicity cloud, with seismic events located at sites where previous experiments (i.e., experiments HS8, HS4) had already induced seismicity, possibly indicating interaction with the HS8 and HS4 stimulated zones. Thus, no main cloud with a distinct orientation could be identified for experiment HF3.

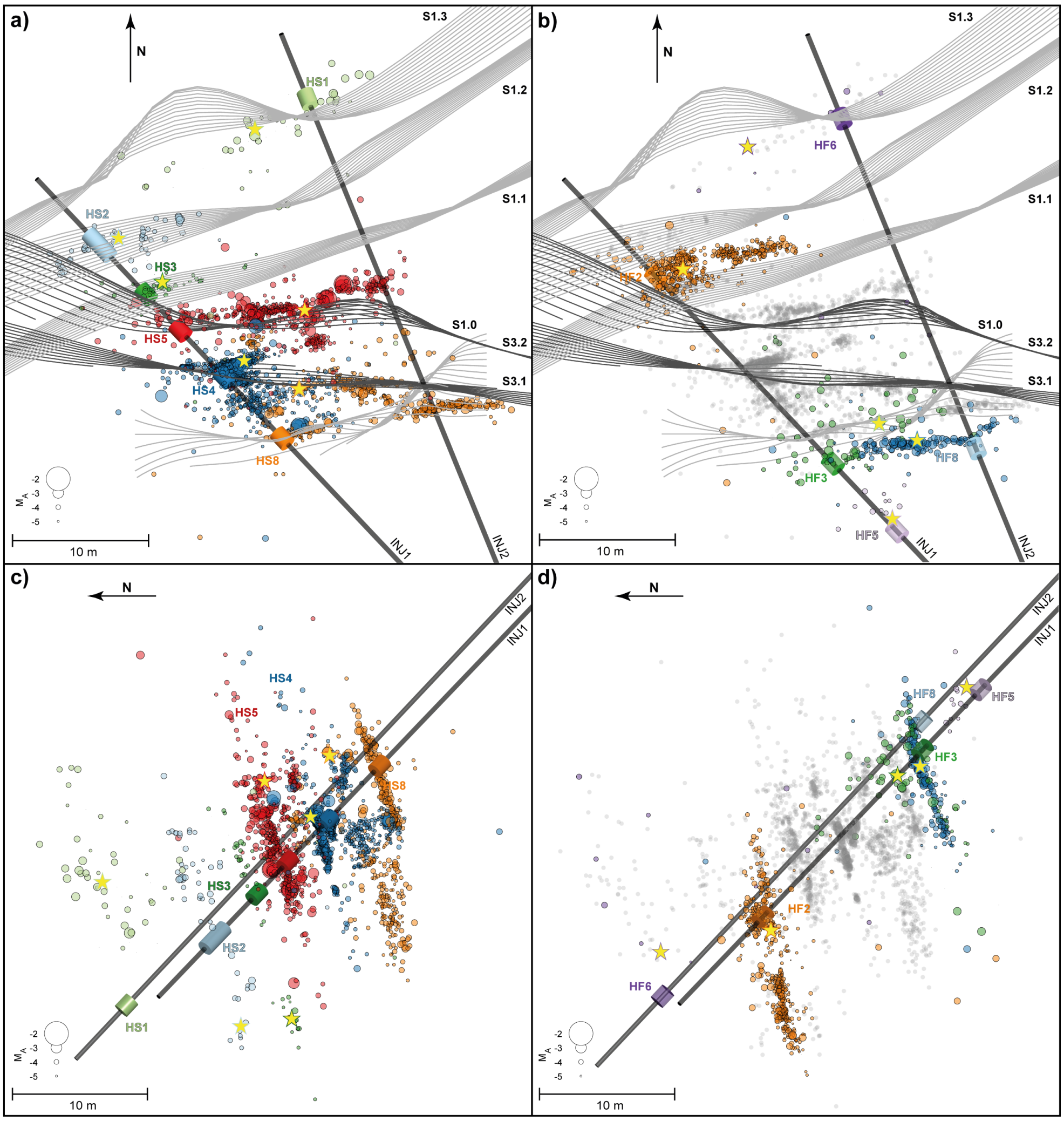

**Figure 6: a) Overview of HS event locations in top view including interpolated shear zones, c) East view and b) overview of HF event locations in top view including interpolated shear zones and d) East view. Injection intervals experiment-wise are color coded. The maximum magnitude of each stimulation experiment is indicated with a yellow star. Note also that in order to improve visibility, the diameter of the injection intervals is exaggerated.**

Planes fitted through the seismic event clouds by orthogonal distance regression are shown in Figure 7 as half circles and their poles in lower hemisphere stereographic projections. The standard deviation of orthogonal dis-

tances of the seismic event locations to the fitted planes is below ±1m, except for experiment HS1 (standard deviation ±1.4 m). The poor plane-fit quality for HS1 events may be associated with increased location uncertainty at the bottom of injection borehole INJ2 (see section 2.2.3).

For injections HS1, HS2, HS3, HS5, HF5 and HF8, fitting a single plane proved to be sufficient; three were observed in HS4, two in experiment HS8 and in experiment HF2 two seismic clusters were observed and planes were fitted for each of these clusters (Figure 7). No plane was fitted for experiment HF3 due to the dispersed character of its seismicity cloud. For experiment HF6, there were too few located seismic events (details of the fitted planes can be found in the supplementary material SM4).

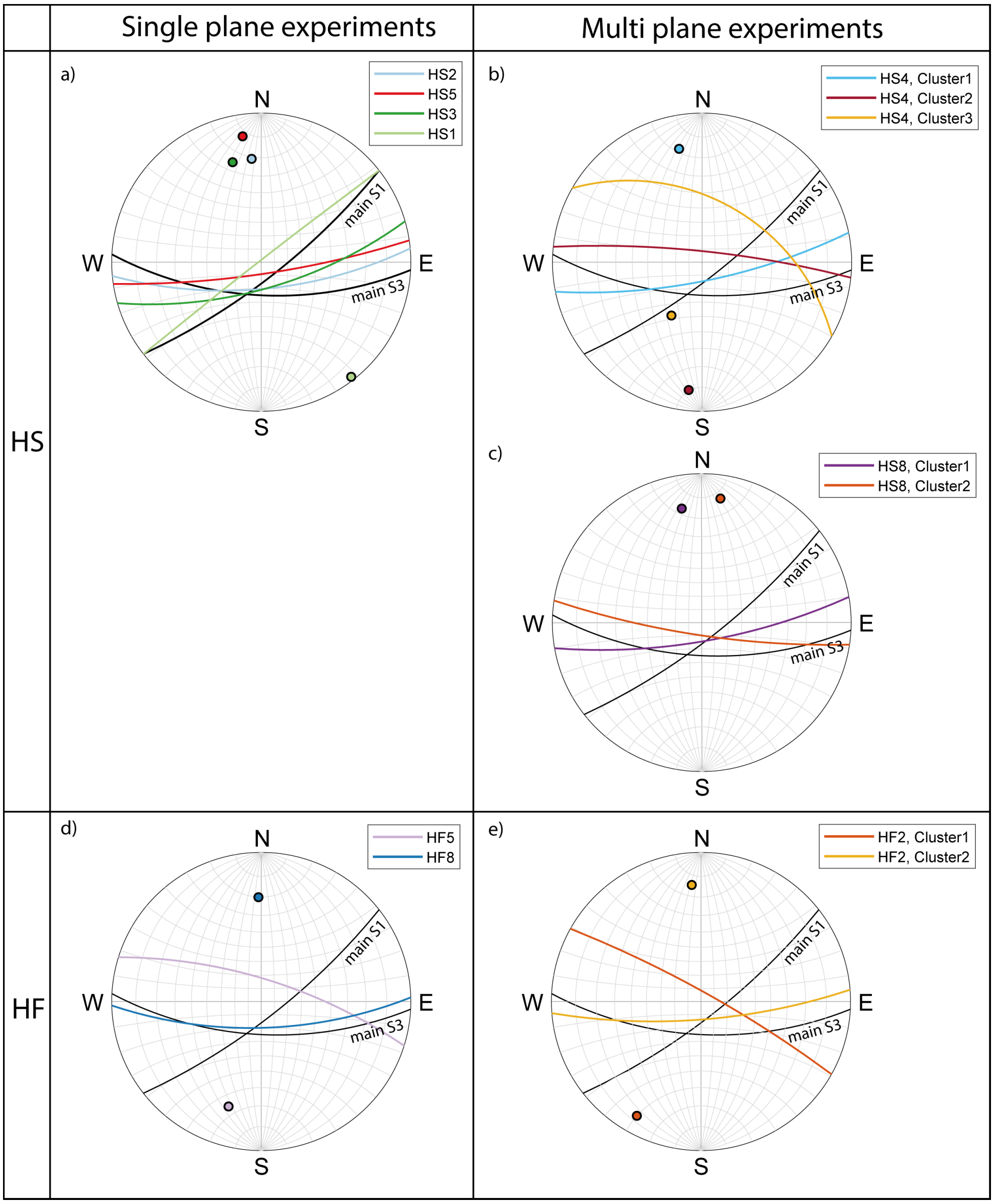

**Figure 7: Orientation of fitted planes and corresponding pole points through seismic clouds in lower hemisphere stereographic plots, including main orientations of shear zone S1 and S3 observed in the tunnels. a) for HS1, 2, 3 and HS5 for which a single planar orientation of stimulation was identified, b) for the three visually identified seismic clusters of injection HS4, c) for the two clusters of injection experiment HS8 d) for HF5 and HF8, for which a single planar orientation of stimulation was identified and e) for the two visually identified seismic clusters of injection HF2.**

Also included in Figure 7 are the main orientations of the S1 and S3 shear zones observed in the surrounding tunnels (Krietsch et al., 2018a). Interestingly, the seismicity clouds of experiments HS2 and HS3, both targeting S1 structures, have an orientation similar to HS5 and to the main orientation of the S3 shear zones. Only the seismicity cloud of the S1 stimulation HS1 is oriented similar to the main orientation of S1 shear zones, although its dip is slightly steeper. The HS4 seismicity produced three distinct cluster orientations: Cluster 1 formed from the injection interval and propagates sub-vertically in the ENE direction, Cluster 2 formed higher up in the injection interval and was oriented EW, parallel to the shear zone S3.1, and Cluster 3 is a new fracture that formed during the main stimulation cycle (C3). The fracture formed at a location that was deemed to be fracture-free during geological characterization prior to the stimulation experiments. In addition, the formation of the new fracture was observed as a strong and abrupt opening by a 1 m long strain monitoring sensor installed in a borehole (i.e., FBS2 see also Figure 2) parallel to the S3.1 shear zone (Figure 8d). For more information about the strain monitoring system see Doetsch et al. (2018a) and Krietsch et al. (in preparation-b). The strong tensile signal from the strain monitoring interval at the 24 m borehole depth and the contraction of the adjacent strain monitoring intervals began when there was a step rate increase of fluid flow. The opening character lasted for about 10 minutes and was accompanied by the HS4 seismicity Cluster 3. Peak extensional strain occurred at shut-in. Contraction of the fracture during the shut-in phase is also associated with seismicity, after both cycles 3 and 4.

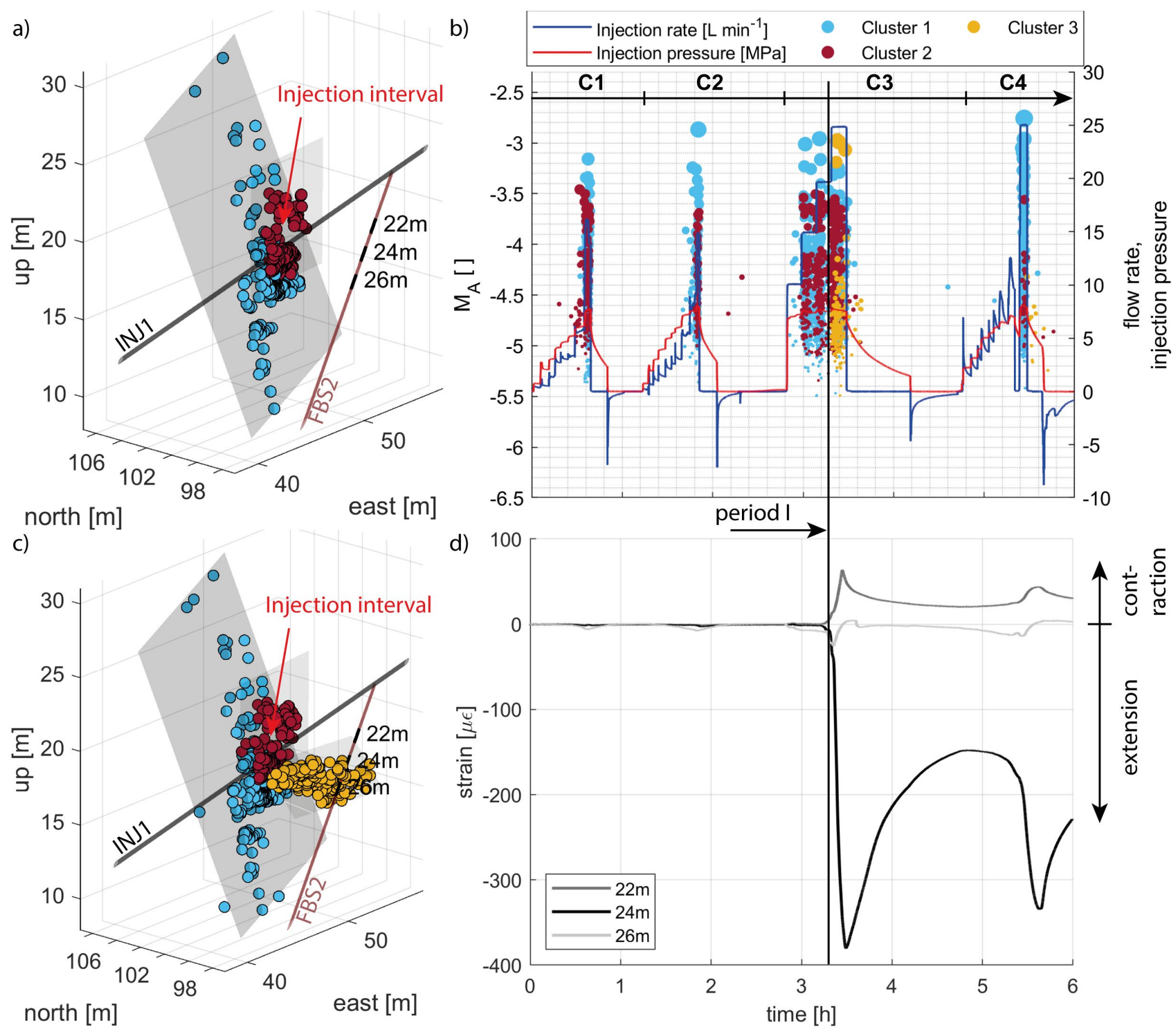

**Figure 8: Observation of newly formed fracture during injection experiment HS4. a) the spatial distribution of seismicity clusters observed during period I, color-coded according to cluster affinity, along with injection borehole INJ1 and the strain monitoring intervals at 22, 24 and 26 m in the strain monitoring borehole FBS1. b) the temporal evolution of seismicity including injection parameters. c) spatial distribution of all seismicity of the three main clusters. d) the strain evolution of strain monitoring intervals at the specified depths.**

The two clusters of experiment HS8 indicate an initial stimulation of shear zone S1.0 in the ENE direction, hydraulically connecting the injection interval with injection borehole INJ2. The second seismicity cluster indicates stimulation along lower regions of shear zone S3.1 in the EW direction, possibly because the zone stimulated during experiment HS4 was reactivated during HS8.

The seismicity cloud from experiment HF8 is oriented EW, again comparable to the orientation of S3, while the seismicity cloud of HF5 deviates from this orientation. Experiment HF2 contains two main seismicity clusters: Cluster 1 includes the events propagating from the injection interval and is oriented comparable to the orientation of HF5. With ongoing stimulation, Cluster 2 is formed and orients itself in the E-W direction.

### 4.3 Propagation of seismicity

Investigating 1D propagation of seismic events reveals a maximum distance of 20 m between the injection interval and seismic events of a respective injection experiment (Figure 9). Diffusivity values were derived from time-distance representations of seismicity based on the concept of seismic triggering fronts in a homogeneous, isotropic

and poroelastic medium introduced by Shapiro et al. (2002). The distance of the seismicity front to the injection interval is $r = \sqrt{4\pi Dt}$, where $D$ is the scalar hydraulic diffusivity in $\frac{m^2}{s}$ and $t$ is the time from the beginning of injection. The time from the beginning of injection was substituted with $t = \frac{\Delta V_{cycle}}{mean(Q)_{cylce}}$, where $\Delta V_{cycle}$ is the injected cumulative volume per cycle and $mean(Q)_{cylce}$ represents the mean flow rate of the cycle (see as an example experiment HS5 in Figure 9a). Diffusivity values over all injection experiments are in the range of 1e-3 to 1e-2 $m^2s^{-1}$, whereby S1 stimulation experiments tend to show higher diffusivity values. For experiments targeting S1 shear zones, located events in the early cycles (C1, C2) cover more than 80 % of the maximum distance to the injection interval.

Systematic seismic propagation patterns are difficult to derive from the 1D representations (Figure 9a); we further investigated the 2D seismicity propagation along the reactivated fracture zones, which allows for an estimation of the seismically activated area during each injection experiment. Thus, seismic event locations for each experiment were projected onto the best fitting planes.

The upper bound of the seismically activated area was estimated based on boundary edges (convex hull) surrounding the collapsed seismic events (Barber et al., 1996). For the lower bound, the seismically activated area was inferred using an estimate based on the concave hull after Gurram et al. (2007) using a lambda parameter of 0.5. The largest area (convex hull) was activated during injection experiment HS5 and amounts in almost 300m2 (see also supplementary material SM5).

In general, only a few experiments (e.g., HS8 and HS4) show concentric growth of seismicity. Seismicity of subsequent cycles often occurs at the same location, which suggests that the same fracture zones are reactivated during repeated injection. Furthermore, the seismicity of many of the injection experiments shows a change in propagation direction for repeated cycles (HS1, HS2, HS3 and HS5). As an example, the convex hull of the consecutive seismically activated area over cycles for the injection experiment HS5 is shown in Figure 9, b-d. During this experiment, the injection interval is hydraulically connected to injection borehole INJ2, and is accompanied by a seismicity cloud moving towards injection borehole INJ2. However, initially (i.e. during cycles 1 and 2) seismicity propagates upwards and only changes direction towards INJ2 in the subsequent cycles.

Also included in Figure 9 b-d are the seismic velocity changes derived with 4D seismic tomography outlined by Schopper et al. (under review), which are associated with pressure and effective stress changes (Doetsch et al., 2018b). The 4D velocity variations are projected onto the best fitting plane of the respective experiment. The propagation of the seismicity cloud towards injection borehole INJ2 is supported by negative seismic velocity variations compared to the initial velocity field at the beginning of the experiment (Figure 9, b-d). However, the seismicity clouds do not consistently propagate along with the seismic velocity decreases, which indicates that seismicity clouds may not adequately represent the pressurized zones.

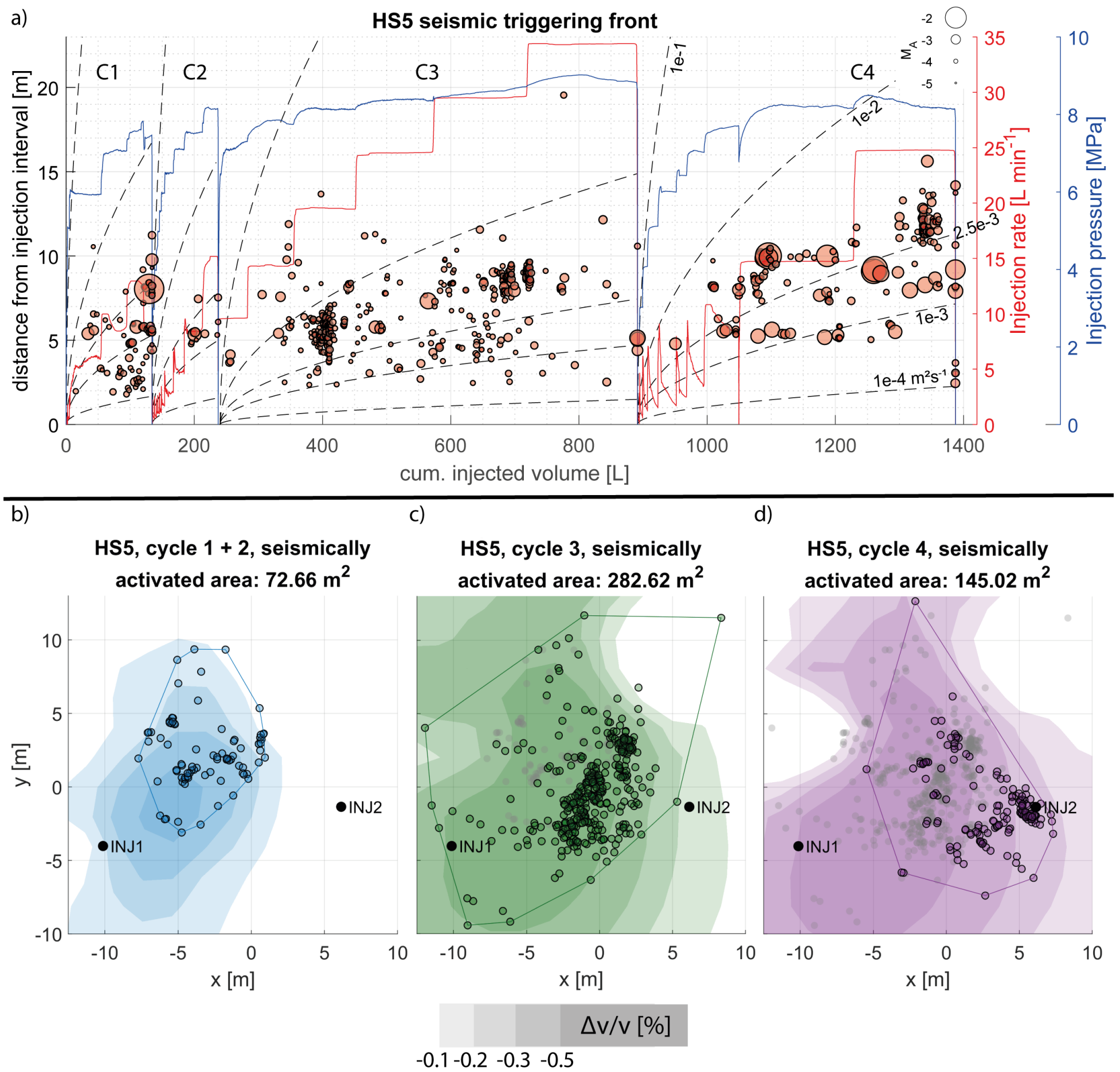

**Figure 9: a) Distance of induced events to the injection interval as a function of cumulative injected volume with an estimate of the triggering front in a homogeneous, isotropic and poroelastic medium with varying hydraulic diffusivity values (1e-1, 1e-2, 2.5e-3, 1e-3, 1e-4 m²/s) along with injection parameters (injection rate, injection pressure) of injection experiment HF2. The figures for the remaining injection experiments can be found in the supplementary material SM6. b)-d) Example of seismicity superimposed with velocity variations on best-fitting plane throughout the four injection cycles of experiment HS5, along with the location of piercing of injection borehole INJ1 and INJ2. The seismically activated area represents the convex hull. Seismic events of previous cycles are colored in grey in subsequent plots. Velocity variations represent the change to the velocity field at the beginning of the experiment and are taken during peak injection of the respective cycle. Regions with velocity variations smaller than -0.1 compared to the velocity field at the beginning of the experiment are colored in maximum transparency. Minimum transparency represents a velocity change of -0.5 compared to the velocity field inferred at the beginning of the experiment.**

### 4.4 Frequency magnitude distributions

The Gutenberg-Richter a- and b-values are estimated for partial catalogs of respective injection experiments defined by the magnitude of completeness, $M_C$. The latter was determined per experiment using the goodness of fit method introduced by Wiemer and Wyss (2000). a-, b-values and its uncertainty are calculated using the modified Maximum Likelihood technique published by Marzocchi and Sandri (2009). a-values are normalized by the injected volume to derive the so-called seismogenic index, Σ (Dinske & Shapiro, 2013). Figure 10 shows frequency magnitude distributions (FMD) of all injection experiments. $M_C$, and with it a- and b-values, were estimated for injection experiments exhibiting more than 20 seismic events and a goodness of fit quality of more than 90 %. Exceptions were made for injection experiment HS3 and Cluster 3 of experiment HS4, where the goodness of fit quality lies above 85 %. $M_C$ is lowest for injections in the focal point of the seismic network (HS4: -4.90, HS5: -4.80, HF2: -4.78). For injection experiment HS4, a bimodal frequency magnitude distribution was observed. For a-, and b-value calculations, the higher $M_C$ of -4.32 was used. $M_C$ increases for injections performed outside the network focus (HS3: -4.66, HS2: -4.39, HS8: -4.38) and is highest for the injection experiments performed towards the bottom of injection borehole two (HS1: -4.05) and towards the tops of the two injection boreholes (HF3: -4.14, HF8: -4.02). Thus, for these experiments the range between the maximum induced magnitude and $M_C$ is small.

The HS injection experiments (Figure 10a) targeting S1 shear zones exhibited larger b-values (HS1: 1.93±0.39, HS2: 1.69±0.26, HS3: 1.93±0.37) and lower seismogenic indices (HS1: -6.6, HS2: -5.8, HS3: -7.6) compared to the b-values of injections into S3 shear zones (HS4: 1.36±0.04, HS5: 1.03±0.05) with higher seismogenic indices (HS4: -3.0, HS5: -2.4). Again, HS8 - an injection into the S1 shear zone south of S3 with migration of seismicity into the S3 shear zone - forms an intermediate case between injections into S1 and S3 with a b-value of 1.61±0.12 and a seismogenic index of -4.9. The b-value for the bimodal FMD of injection HS4 in a magnitude range of -4.9 to -4.35 lies below 1 as compared to 1.36 above magnitude -4.35.

The b-value for the HF2 experiment (Figure 10b) north of the S3 shear zones is comparatively low at 1.35±0.08, with a seismogenic index of -4.0. Experiments HF3 and HF8 south of the S3 shear zones at a similar depth of injection borehole INJ1 and INJ2, respectively, exhibited b-values of 1.55±0.26 and 2.66±0.36. Seismogenic indices for the two injection experiments were -4.8 for HF3 and -9.0 for injection HF8.

A more detailed analysis of the bimodal FMD of HS4 reveals that the bimodal character does not disappear if the FMD is split up into all four injection cycles (Figure 10c). Also for FMDs of individual seismicity clusters (see section 3.1), the seismicity cluster closest to the metabasic dykes (Cluster 1) confirms the bimodal characteristic (Figure 10d). The cloud subparallel to the metabasic dyke (Cluster 2) shows a bimodal character, but with a break in scaling at higher magnitudes compared to the FMD of Cluster 1. The new fracture induced and propagated during injection cycle 3 (Cluster 3) does not show the bimodal characteristic, but reveals five higher magnitude events than would be expected.

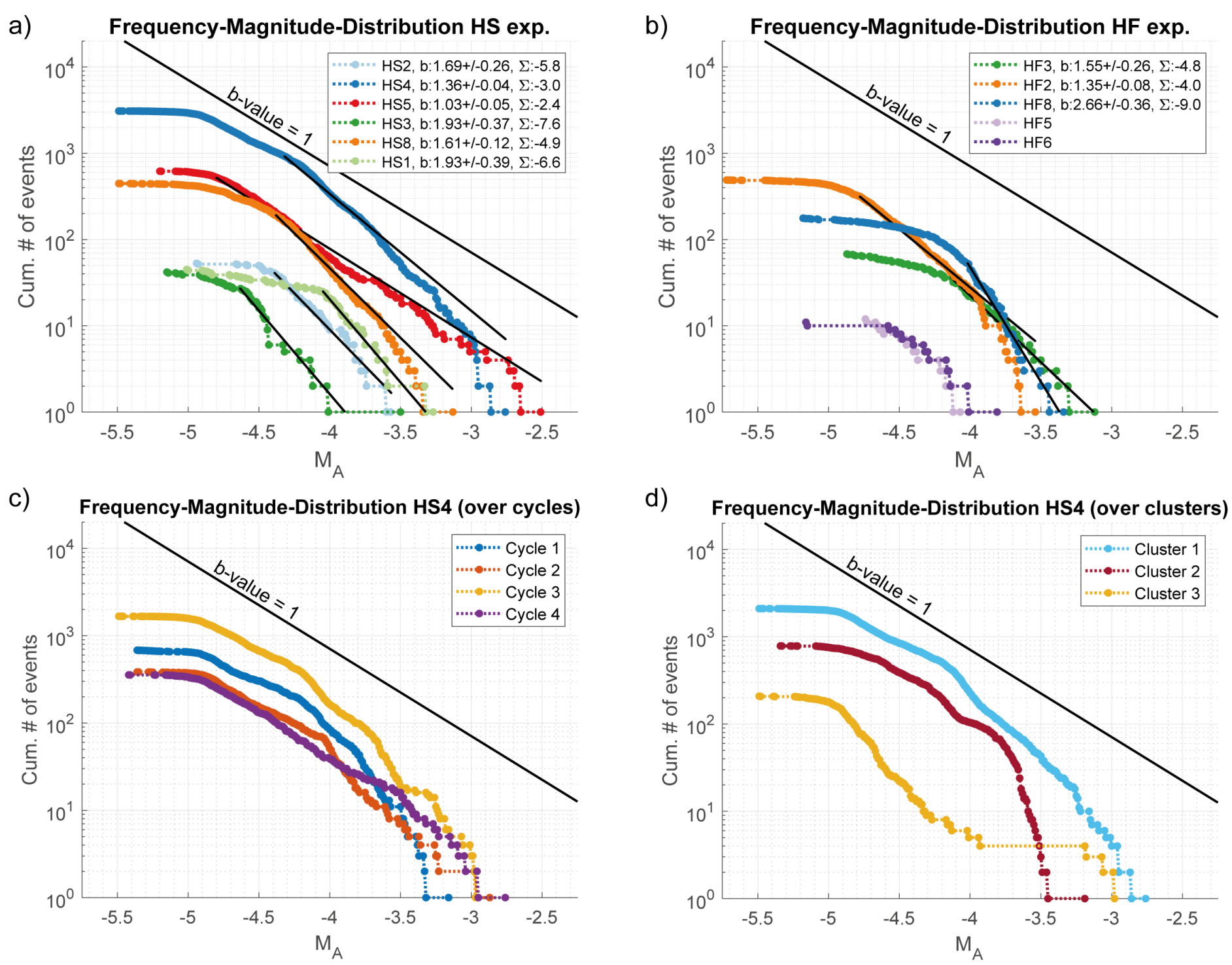

735 **Figure 10: Frequency magnitude distributions for the HS (a) and the HF (b) injection experiments along with estimated $M_C$, b-values and seismogenic indices. Injection experiments in legends are ordered in a chronological manner, whereby HS injection experiments were performed in February 2017 and HF injection experiments were executed in Mai 2017. Frequency magnitude distributions for injection experiment HS4, resolved in c) injection cycles (Cycle 1 – Cycle 4), and d) clusters, introduced in section 3.1, Figure 7b and Figure 8. Uncertainties in b-values are estimated after Marzocchi and Sandri (2009)**

### 4.5 Maximum observed magnitude vs. stimulated area, a-, b-values

The maximum observed magnitudes per injection experiments ranged over 1.5 magnitudes. The observed maximum magnitudes showed only a slight tendency to increase as the injected fluid volumes increased (Figure 1), possibly owing to the fact that the injected volumes were only marginally different (900 – 1500 l). However, a stronger relationship was seen between maximum observed seismic magnitudes and the seismically activated area (Figure 12). Injection experiment HS5 represents the highest magnitude event as well as the largest seismically activated area (285 $m^2$). Also during injection experiment HS4 in which several planes were seismically activated resulting in a large seismically activated area, a rather large magnitude seismic event was induced. There were no obvious differences in the maximum induced magnitude in relation to injected volume or seismically activated area between the HS and HF injection experiments.

Gutenberg-Richter b-values and seismogenic indices show a high variability, but no correlation with the seismically activated area (Figure 11). Nonetheless, injection experiment HS5, during which the largest area was activated and the largest magnitude event was induced, also shows the lowest b-value and the highest seismic productivity. A comparatively small area was activated during injection experiment HF2 with similar low b-values and high seismogenic indices.

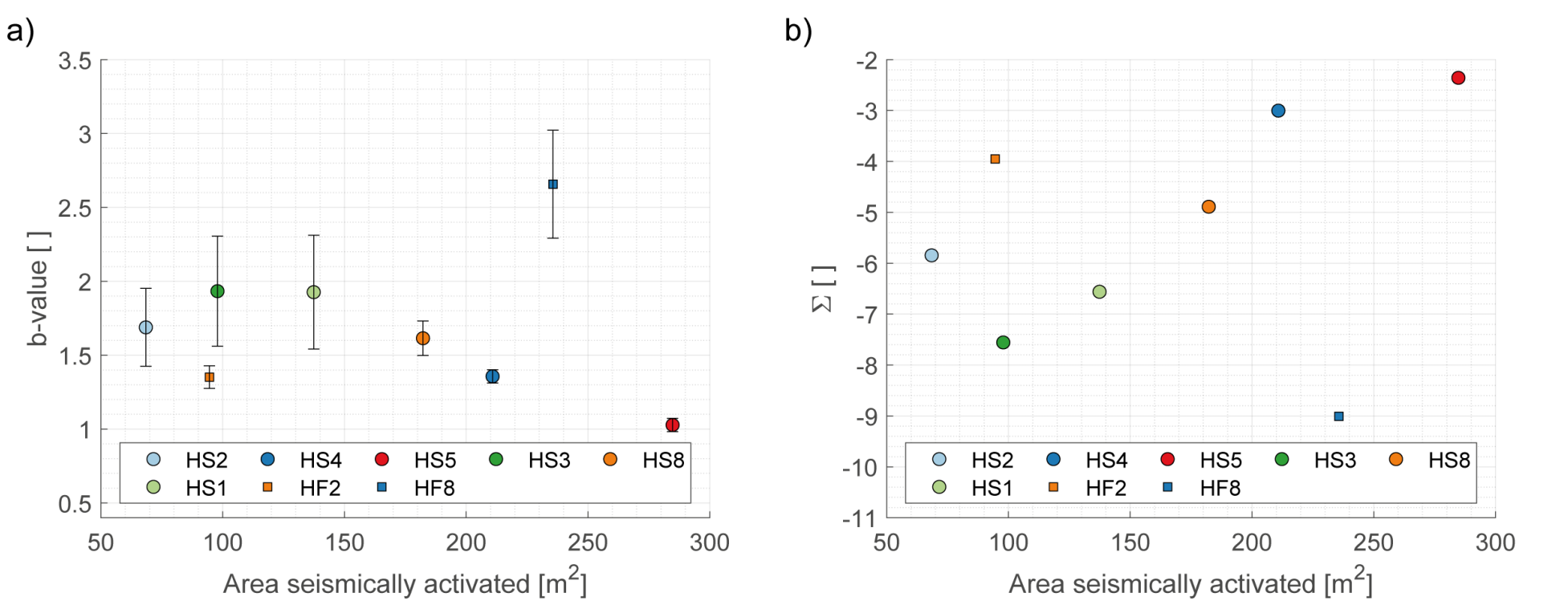

**Figure 11: a) b-values along with uncertainties plotted against seismically activated area and b) seismogenic indices plotted against seismically activated area from experiments for which the b-values, seismogenic indices and area could be estimated.**

### 4.6 Seismic injection efficiency, ratio of seismic/aseismic deformation

In the following, we estimate the *seismic moment release* (referred to as *$M_0$ seismic* in Figure 12b) and compare it with the *total moment released* by stimulation *(M0 displacement)* as well as with a quantity termed *hydraulic moment release (M0 hydraulic)*. The lower bound estimate of seismic moment release during each injection experiment was determined by adding up the seismic moment of each located seismic event during the respective injection experiment. In order to estimate the experimental specific upper bound of the seismic moment release, the Gutenberg-Richter a- and b-values, determined in section 3.5, were used to extrapolate the seismicity rates down to a magnitude of -9. Such small magnitudes were observed on the laboratory scale by Selvadurai (2019). Also McLaskey and Lockner (2014) and Yoshimitsu et al. (2014) observed very small magnitudes (i.e., M -7) and self-similarity down to these magnitudes. In situ, Goodfellow and Young (2014) observed magnitudes down to -7.5. The cumulative seismic moment release was then computed based on seismicity catalogues expanded by extrapolation to lower magnitudes. For a best guess estimate of the seismic moment release, magnitudes down to

a minimum magnitude of -6 were included (symbols in Figure 12a). A high range of possible seismic moment release was observed for injection experiments with high b-values (i.e., HS1, HS3, HF8), because the small magnitude seismic events strongly contribute to the cumulative seismic moment release. Assuming the best guess scenario, a single earthquake per injection experiment would have induced a moment magnitude MW in the range of -3 to -1. Assuming a stress drop of 1MPa or 0.1MPa, respectively, and a source model by Brune (1970), this would correspond to a source radius of 0.3 - 2.2 m / 0.6 – 4.8 m and a ruptured area of 0.26 – 15.5 m2 / 0.28 - 18 m2).

The *equivalent hydraulic moment* was calculated from the determined hydraulic injection energy. The hydraulic injection energy was estimated using $E_{hyd} = \int pQdt$, where $p$ is the injection pressure and $Q$ is the injection flow rate that are both integrated over the entire injection time. The pumped hydraulic energy is then converted to an equivalent seismic moment using $M_0 = \frac{\mu}{\Delta\sigma} E_{hyd}$ (Aki & Richards, 2002; De Barros et al., 2019) where $\mu$ is the shear modulus, chosen to be 30GPa and $\Delta\sigma$ represents the static stress drop assumed to be between 1MPa and 0.1MPa. The best guess estimate represents the equivalent seismic moment averaging the aforementioned stress drop range.

The *total moment released by stimulation* can be estimated from borehole dislocations in the injection interval (Figure 12c), that was determined from acoustic televiewer (ATV) measurements before and after each injection experiment (i.e., for injection experiment HS2: 0.95 mm, for HS4: 0.95mm, for HS3: 1.25mm, for HS8: 0.45mm and for HS1: 0.75 mm, see Krietsch et al. (in preparation-a)). Note that this is only possible for HS experiments, since in the HF experiments no fault dislocations were observed (Dutler et al., 2019). For the estimate of the seismic moment from the measured displacements at the injection interval, we used $M_0 = \mu AD$, where $\mu$ is the shear modulus, again chosen to be 30GPa, $A$ is the seismically activated area determined in section 3.1 and $D$ is the average slip on the area of rupture. For a lower bound estimate, we assume that an average slip over the entire lower bound seismically activated area (i.e. the concave hull area, see section 3.1) is 10% of the observed slip at the injection interval. For the upper bound estimate, we assume that the average slip across the entire upper bound seismically activated area (i.e., the convex hull area estimate) corresponds to 50% of the observed slip at injection intervals. 25% of the observed slip as well 50 % of the estimated seismically activated area were used for the best guess estimate (symbols in Figure 12c).

To estimate seismic injection efficiencies (i.e. the ratio between seismic moment released to equivalent hydraulic moment, Figure 12d) and the ratio between seismic and total deformation, the best guess estimates of the equivalent hydraulic and displacement moment were used. The cumulative seismic moment release was varied according to the minimum magnitude at which seismicity rates were extrapolated. When integrating to a minimum magnitude of -6, seismic injection efficiencies lie in the range of 1.9 x 10-6 (HS3) and 5 x 10-4 (HF8); injection experiment HS4 showed a high value of 1 x 10-4 with minor changes as the integration magnitude decreased, due to the low b-value (i.e. due to the small contribution of small magnitude events to the cumulative seismic moment). Seismic injection efficiencies (excluding experiment HF8) tended to converge to a value in the range of 1.6 x 10-5 (HF2) and 3.2 x 10-4 (HS1) when integrating to a minimum magnitude of -9.

The ratio between seismic and total moment release (Figure 12e), considering events with magnitudes down to -6, ranged from 6 x 10-4 (HS3) to 6 x 10-2 (HS4). Integrating the seismic moment to a minimum magnitude of -9

leads to a convergence of the ratio between seismic and total deformation to values of 1.3 x 10-3 (HS3) to 1.8 x 10-2 (HS1).

We emphasize that the cumulative seismic moment, the equivalent hydraulic moment and the equivalent total moment from dislocation observations, are prone to a high level of uncertainty. Thus, uncertainties in the seismic injection efficiencies and the ratio between seismic and total moment give only crude estimates with uncertainties that possibly exceed one order of magnitude.

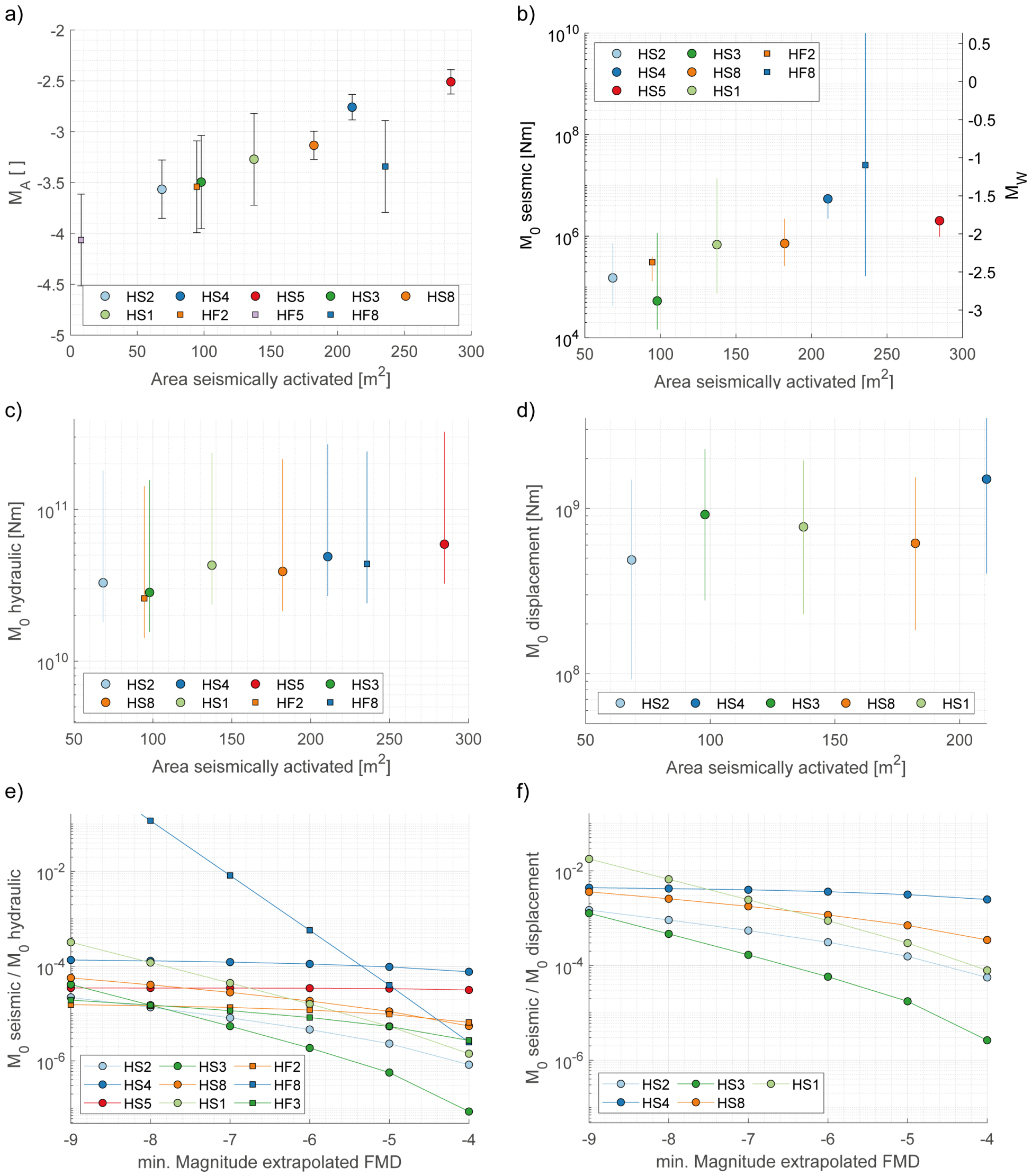

**Figure 12: a) maximum observed magnitudes (error bars represent the standard deviation of all magnitude estimates of the respective event) with respect to seismically activated area (the estimated seismically activated area represents the mean between the upper and lower bound of the area estimate of section 3.1). b) Estimated radiated seismic moment from extrapolated Gutenberg-Richter parameter (upper bound and best guess) and located seismic events (lower**

**bound) along with the equivalent moment magnitude, c) equivalent hydraulic moment estimated from injection parameter (i.e., flow rate, injection pressure), d) equivalent moment estimate from acoustic televiewer displacement measurements at the injection interval, e) seismic injection efficiency against the magnitude level used for seismic moment extrapolation, f) ratio between seismic moment and equivalent seismic moment estimated from displacement measurements against the magnitude level used for seismic moment extrapolation.**

## 4.7 Network performance

During HS and HF injection experiments 5'157 seismic events were located (of a total of 20'557 detections) using 32'529 first arrival P-wave pics. Most first arrivals were recorded with the eight borehole AEs (86%) in close proximity to the injection intervals (distance AEs to injection intervals: 3 – 30 m), a small number of arrivals were recorded with the tunnel AEs (8.6%) and on the four additional borehole AEs installed in SBH4 (4.6%). Only five P-wave arrivals with poor signal to noise ratios (<2dB) were recorded on accelerometers.

The number of both detected and located seismic events varies significantly over the injection experiments. The largest seismic events with magnitudes MA -2.5 – MA -3 were recorded on all AE sensors, i.e. at a source to sensor distance of up to 50 m (Figure 13a, b). The smallest events (< MA -6) were recorded only at distances < 5 m. The detection threshold of low magnitude events (i.e. red lines in Figure 13a) indicate that the sensitivity of the borehole AE sensors (R16 – R23, Figure 3a) improves faster with shorter distances as would be expected from the tunnel sensors. The maximum observed frequency of the recorded seismic signals (i.e., frequency recorded with a spectral S/N > 3dB; not to be confused with the corner frequency) may exceed 100 kHz for events close to AE sensors (< 10 m), but quickly decays with distance (Figure 13c). An example of the corrected displacement spectrum recorded on tunnel AEs is shown in Figure 13b. Fitting the theoretical displacement source spectrum is only possible for a limited frequency range (~1 – 10 kHz). Frequencies below 1 kHz are filtered through hardware high-pass filters. High frequencies are strongly attenuated, possibly due to the relatively low Q-factor.

When investigating spatial properties of seismicity clouds one has to be aware of the spatially varying network sensitivity. Our eight borehole AE sensors close to the injection intervals are conclusive for an increased network sensitivity in the experimental volume close to the injection boreholes. In addition to the source-receiver distance, the sensitivity of the network is significantly influenced by the directivity of the AE sensors, i.e. events with incident angles > 50° in the Grimsel experiment are less likely to be detected as described in subsection 2.2.4. The network sensitivity decreases for experiments performed below the borehole AE sensors, e.g. experiment HS1, which may explain why we only observed seismic events located above the HS1 interval towards the borehole array. Also, for experiments performed in the upper region of the borehole array (HS8, HF3 and HF8), network sensitivity is decreased, although this is less pronounced than for experiment HS1, because of the proximity to the tunnel AE sensors. Network sensitivity also decreases in the NS and EW direction away from the borehole array. The decay of spatial network sensitivity away from the focal point of the borehole AE sensors has a potential impact on observed spatial properties of seismicity clouds. Thus, along with the decreasing network sensitivity, $M_C$ estimates increase as the distance from the focal point of the borehole AE sensors increases (Figure 13d). However, a comparison of the seismicity clouds between experiments is only possible if events with magnitudes > $M_C$ are considered. A more detailed analysis of the network sensitivity may involve 3D mapping of $M_C$ with probability-based $M_C$ estimates (Schorlemmer & Woessner, 2008; Plenkers et al., 2011) or with a Bayesian approach (Mignan et al., 2011) at locations where no seismicity was observed. This, however, is the scope of future work.

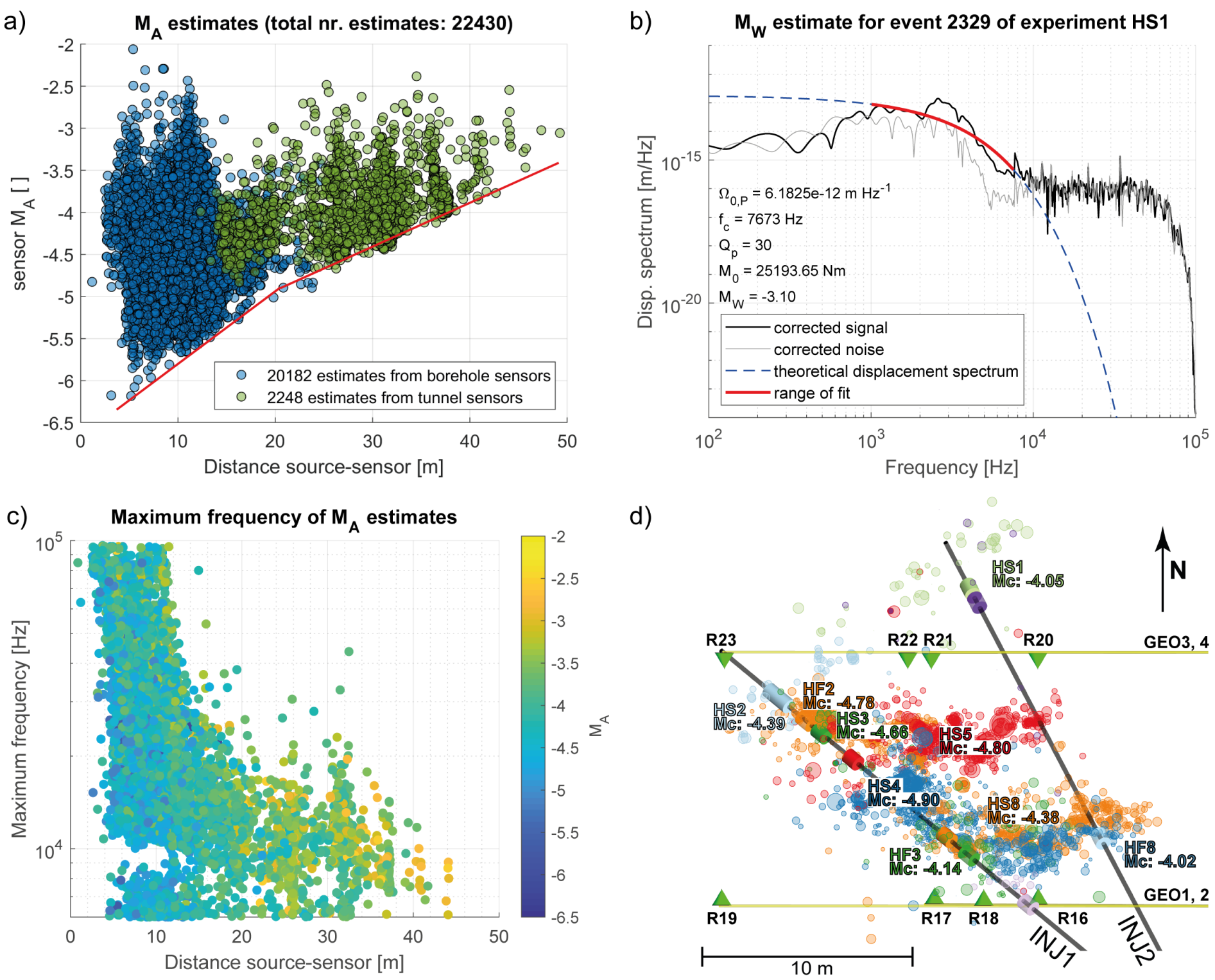

**Figure 13: a) Estimated magnitudes M$_A$ from recorded borehole and tunnel AE sensors vs. source to sensor distance, including network sensitivity lines (red). b) Observed corrected displacement spectrum (black solid line) and noise spectrum (grey solid line) along with fitted theoretical displacement spectrum after Boatwright (1978) for a frequency range from 1 to 8 kHz. c) Maximum frequency vs. event to sensor distance of all seismic events. d) M$_C$ estimates for all experiments, along with located seismicity from all experiments and borehole AE sensors (R16 – R23).**

Another capability of the seismic network, which is to some extent limited, is the ability to record S-waves (as can be seen from the example events in Figure 3). The reasoning for why there are often no clear S-waves observed remains speculative. Clear S-waves have been observed at comparable sites where similar monitoring equipment was installed (Kwiatek et al., 2011; Zang et al., 2016; Dresen et al., 2019). A plausible reason might be that the designed waterproof sensor-shuttles in which the borehole AE sensors were deployed could influence the ability to record S-waves. Also, the water filled boreholes might have an impact on S-wave recording capability. The recordings from AEs installed at the tunnel level might lack S-wave energy because of an alignment of the AEs towards the seismicity occurring around the injection intervals.

# 5 Discussion, conclusions and outlook

The hydraulic stimulation experiments performed at the Grimsel Test Site aimed to investigate the influence of different geological settings (i.e. pre-existing fractures with variable orientation and architecture, HS, intact rock, HF) to high-pressure fluid injection in terms of induced seismicity, permeability increase, pressure propagation and rock deformation. Short borehole intervals of 1 – 2 m length were stimulated with standardized injection protocols - one each for the HS and HF experiments - and a total injected volume of about 1 $m^3$. The injection protocol differed for HS and HF because, during HF experiments, the formation breakdown pressure of the rock had to be overcome for fracture initiation, while shearing during HS experiments can be initiated at pressures below the minimum principle stress. Thus, the HF experiments required higher injection rates and pressures than the HS experiments. It is also important to mention that the HF experiments were conducted in the same rock volume after the HS experiments were completed, which may have already altered the stress conditions in the rock mass. We argue that despite these differences between HS and HF experiments, comparing the process characteristics of all injection experiments is justified.

## 5.1 A highly-variable seismic response and the role of geology

Remarkable is the large variability in the seismic responses between experiments conducted within less than a 25 m borehole length, which is expressed in the wide range of seismogenic indices (-9 to -2) and b-values (1 to 2.7) (Figure 11). The number of detected and located events during a stimulation depends on the detection ability of the sensor network, which is primarily a function of the distance (Mignan et al., 2011). However, even at a homogeneous completeness level of -4.2, the seismic response varies widely (Figure 5c, d). Such variability is comparable to the variability between cases worldwide, involving both projects with predominant HF stimulation in the shale gas context and HS for geothermal exploitation (Figure 14c, (Dinske and Shapiro, 2013;Mignan et al., 2017)). While Dinske and Shapiro (2013) suggest that there is a large difference in the seismic response during HF-dominated stimulations in shale gas projects and HS-dominated stimulations in geothermal applications, a systematic difference between the HS and HF experiments performed in crystalline rock, was not discernible here. Also, the use of the shear thinning xanthan-salt-water mixture during the HF experiments did not have an observable effect on the seismic response. The fact that the HF experiments were conducted in a rock mass where previous HS experiments could potentially have initiated some stress relaxation, may explain the tendency for fewer events during the HF experiments. However, experiment HF6, which can be interpreted as continuation of the HS1 experiment, induced only a few seismic events because the zone was stimulated twice. In contrast, the dispersed character of seismic events in the HF3 experiment may be explained by the interaction of new fractures with the surrounding faults S1.0, S3.1 and S3.2. We conclude that in our experiment HS and HF are similarly seismogenic, because HF strongly interacts with the pre-existing fracture network leading to similar seismic responses as the injections directly into pre-existing fractures.

While differences in the seismic response between HF and HS were not evident, the geological setting seems important for the substantial differences seen in the seismic response in terms of magnitude distributions as well as in terms of orientation and propagation of seismicity. We observed that experiments performed directly on or in the vicinity of the highly fractured brittle-ductile S3 shear zones (Figure 14a, i.e., experiment HS5, HS4 and HS8, HF3 respectively) are characterized by an enhanced seismic response. This observation is in agreement with the hypothesis gained from larger-scale stimulations, which states that well developed brittle fault zones (i.e., connecting fractures that form larger features) lead to a comparatively high seismic moment release in response to

high-pressure fluid injection (McClure & Horne, 2014; De Barros et al., 2016). An exception is experiment HF2, which shows an increased seismic response with possibly no influence from S3 structures. Injection experiment HF2 was performed between the ductile shear zones S1.1 and S1.2, north of shear zone S3. At this location the reactivated structure (i.e., Cluster 1 and Cluster 2 of HF2, see also Figure 7) may support an increased amount of shear stress, which led to an increased seismic response.

Not only do the seismic responses (i.e. b-value and the seismogenic indices) indicate a strong geological influence, but also the seismicity detection rate in relation to injected volume (Figure 5) shows a different seismic footprints for the two shear zone types. For the injection experiments on the ductile shear zones (S1) more than 50% of all detections are made during the injection of the first 100 l of fluid. In contrast, the S3 shear zones experienced a gradual increase of detections with injected fluid volume (Figure 5a).

The spatial distribution and propagation also appear to be affected by the geology. A concentric growth of seismicity clouds was rarely observed, indicating that the spatial fracture zone heterogeneity had a substantial impact. Seismicity clouds of experiments on ductile shear zones S1 show changing propagation directions and a planar character. Comparing the two S3 stimulations (HS4 and HS5), distinct differences in seismicity patterns were observed, even for stimulations within 3 m from each other in similar geological structures. During HS5, propagation directions changed along an extended seismicity cloud (of 16 m diameter) with a clustered character and regions of increased seismic event density. During the HS4 experiment the seismicity was mostly limited to patches/clusters within a 9 m radius from the injection interval, but with a complex 3D and non-planar architecture (Figure 6, 8, 9).

Beside their tendency of being very seismogenic, the highly fractured S3 shear zones stand out as being the most hydraulically conductive structures in the experimental volume compared to the less conductive S1 shear zones (see injectivities of HS4 and HS5 intervals in Figure 14b). Injectivities at these intervals only increased marginally during stimulation. On the contrary, injectivities for the S1 stimulation experiments on the ductile shear zones and in the intact intervals increased by 2 – 3 orders of magnitude. Again, these observations agree with cases in the literature, for which the most permeable fractures were also found to be the most critically stressed and thus the most seismogenic zones (e.g., Barton et al. (1988);Barton et al. (1995);Barton and Zoback (1998);Evans et al. (2005);Davatzes and Hickman (2010);Baisch et al. (2015);Evans et al. (2015)).

It is also noteworthy that the injectivities for all experiments performed at the brittle-ductile shear zones, the ductile shear zones and in the intact intervals end up in the same order of magnitude (Figure 14b). While initial injectivities are highly dependent on the local geology, final injectivities are very similar (and transmissivities, Brixel et al. (under review)).

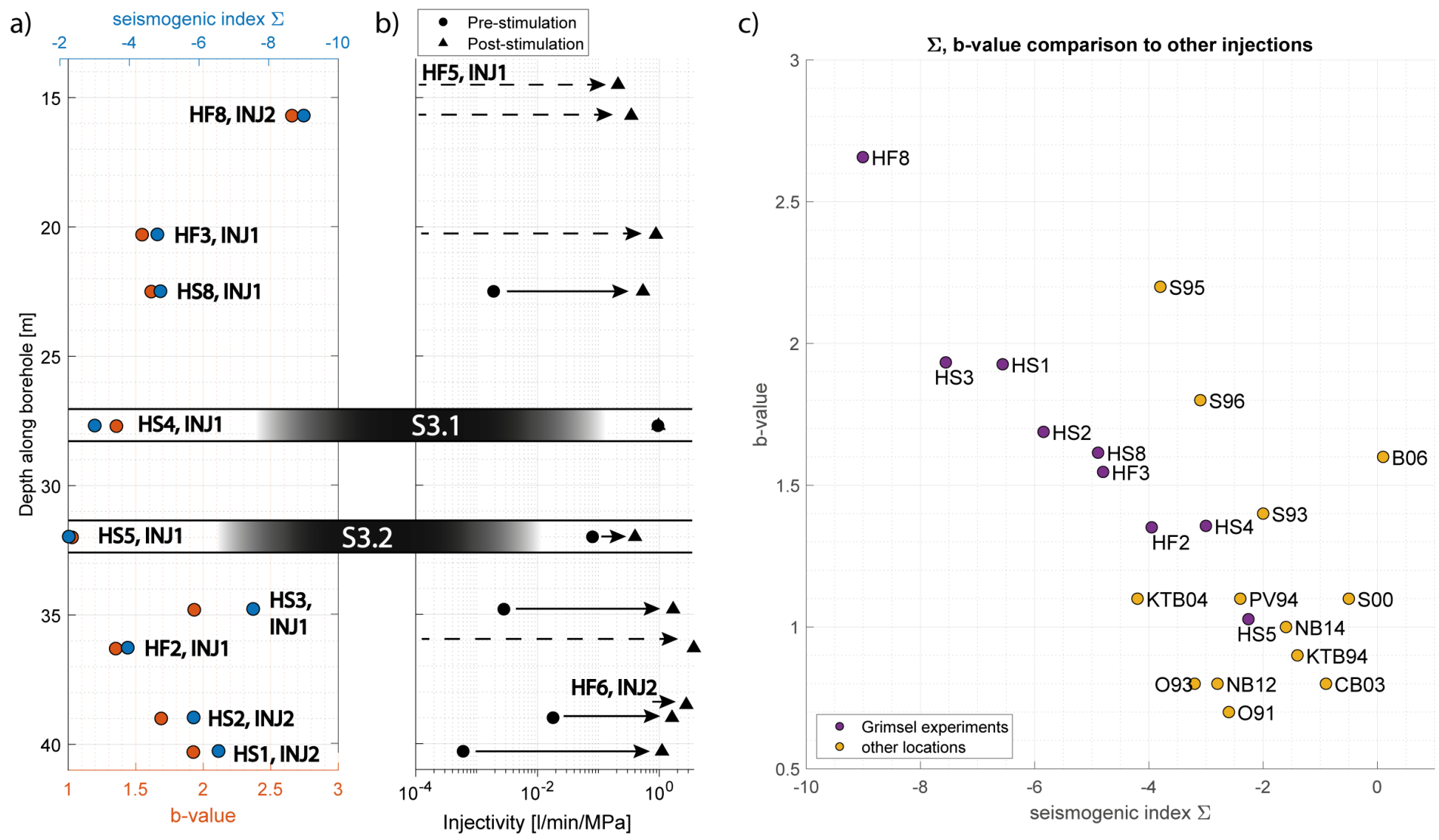

**Figure 14: a) Seismic responses (seismogenic indexes, b-values) along with b) pre- and post-injectivity values of experiments along the depth of the injection boreholes. Location of S3 shear zones and experiment HS4 and HS5 therein are highlighted. Injectivity values of experiment HF5, and HF6 for which the number of located seismic events renders a determination of b-value and seismogenic index impossible, are also included. b) b-values and seismogenic indexes of various high-pressure fluid injections at different sites (source seismogenic indexes, b-values from other locations: Dinske and Shapiro (2013), Shapiro et al. (2013) and Mignan et al. (2017))**

With the aforementioned observations in mind, it is possible to imagine what would have happened if a large open-hole stimulation would have been conducted in INJ1 and IN2, as it was done in most of the previous EGS projects (e.g., Basel, Häring et al. (2008); Soultz, Evans et al. (2005)), instead of several stimulations at selected short intervals. Because of their high transmissivity, flow would have preferentially entered the shear zones S3.1 and S3.2 leading to induced seismicity, mostly dominated by the seismogenic properties of these structures. The result would have been a very limited transmissivity increase together with a strong seismic response. Thus, for larger-scale EGS stimulations, it appears quite promising to selectively stimulate multiple short borehole intervals with comparatively small fluid volumes (i.e. zonal isolation, Meier et al. (2015), during which the transmissivity of low-transmissive structures would be strongly enhanced, while stimulations in intervals at seismogenic fault zones should be avoided if possible. Of course, hydraulic stimulation of short intervals could also be combined with alternative injection schemes (such as described by Zang et al. (2017)). However, the pronounced influence of geology on the aforementioned stimulation parameters in our experiments may imply that the impact of alternative injection strategies on induced seismicity (such as those discussed and proposed in the literature by McClure and Horne (2011);Zimmermann et al. (2014);McClure et al. (2016);Zang et al. (2018)) is limited, since their effects are unlikely to emerge above the strong variability of orders of magnitudes imposed by the geological conditions.

## 5.2 Impact of the stress field

Compared to the observed main orientation of the S1 (NE - SW) and the S3 (EW) shear zones in the tunnels surrounding the experimental volume, the orientation of individual fractures within the S1 and S3 fault zones do show a similar NNW orientation. Also, the orientation of fractures found in the host rock are predominantly NNW

in orientation with some random joint orientations. We combine the orientation of these pre-existing fractures with the slip tendencies inferred from the stress conditions measured 30 m south of shear zone S3.1 (i.e., the unperturbed stress state) and the stress conditions measured in borehole SBH4 (Figure 2) in the vicinity of shear zone S3.1 (i.e., the perturbed stress state). It can be seen that there is an increased susceptibility for the S1 and S3 structures to slip (Figure 15a, b, see also Krietsch et al. (2018a)), when considering the perturbed stress state.

By including both the inferred orientation of the seismicity clouds or their clusters resulting from the injection experiments performed on the shear zones (i.e., the HS experiments) and the stress field, the combined influence of geology and stress field becomes evident. The predominant orientation of seismicity clouds is EW, in agreement with the orientation of pre-existing fractures. Surprisingly, the predominant orientation also holds for the S1 stimulation experiments, even though the main orientation of the S1 shear-zones is NE – SW. Only the seismicity cloud of injection experiment HS1 is oriented in the main S1 direction. However, the orientation of seismicity in EW direction, also for S1 experiments, is not surprising when considering the fracture inventory of the experimental volume and the overlapping pole points of S1 and S3 structures, as well as the increased fracture density with the same orientation (Figure 15a - d). The new fracture created during experiment HS4 (Cluster 3) orients in a direction perpendicular to the minimum principal stress of the perturbed stress field (Villiger et al., 2019). We suggest that the opening of this fracture is due to shear dislocation on shear zone S3.1 (Jung, 2013b).

The orientation of newly initiated fractures in experiment HF5 and the initial fracture (Cluster 1) of experiment HF2 are oriented perpendicular to the minimum principal stress of the perturbed stress state where directional geological features are sparse. We argue that in these two cases, new fractures were created that propagated perpendicular to local minimum principle stress that is well represented by the perturbed stress field. Cluster 2 of experiment HF2 formed at a later time compared to Cluster 1; it possibly formed because of leak-off of fluids through Cluster 1 to the formation. The associated reduction in pore pressure through Cluster 1 suggests a geology-dominated E-W orientation of the seismicity cloud of Cluster 2. Note also, that injection experiment HF2 is the most seismogenic of all the HF experiments. Injection experiment HF8 possibly reactivates pre-existing fractures that were likely optimally-oriented in the local stress field (again best represented by the perturbed stress field).

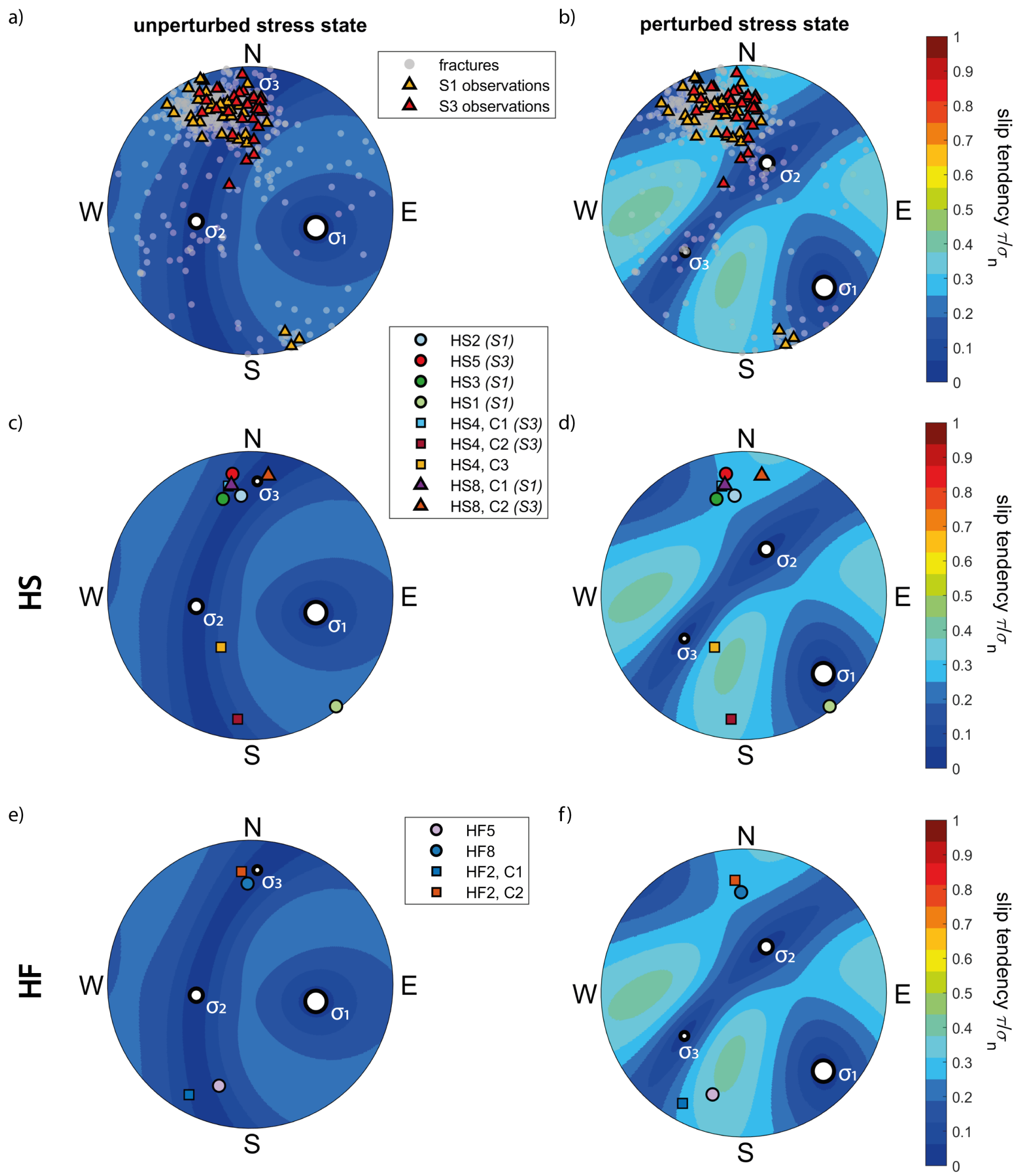

**Figure 15: Principal stress directions of the unperturbed (σ1 = 13.1 MPa, 104° dip direction / 39° dip; σ2 = 9.2 MPa (259°/48°); σ3 = 8.7 MPa (4°/13°)) and perturbed stress state (σ1 = 13.1 MPa, 134° dip direction / 14° dip; σ2 = 8.2 MPa (026°/50°); σ3 = 6.5 MPa (235°/36°)) along with slip tendencies determined from the respective stress state in lower hemisphere stereographic plots, along with a), b) the fracture inventory from borehole observations, c), d) pole points of seismicity cloud orientations of HS experiments, their targeted structures and e), f) orientation of seismicity clouds of HF stimulation experiments.**

Finally, given the similar orientations and similar slip tendencies of the S1 and S3 observations in the boreholes we observed that the influence of the stress field on the seismic response of the reactivated fault regions may be marginal. We argue that the fracture density and fault damage zone architecture determines the seismic response of fault stimulation experiments. To shed more light on the interplay between geology and stress state during

hydraulic stimulation treatments, future work will be undertaken to understand the source mechanisms of seismic events in more detail.

### 5.3 Aseismic deformation

Our experiments indicate that deformation in HS experiments (for which a displacement was measured at the injection interval) is to a large extent aseismic (i.e., < 2% seismic). We also observed the tendency that the amount of aseismic deformation is larger for experiments targeting the S1 structures (Figure 12f). These overall values agree with values determined from hydraulic reactivation of a fault zone in limestone on a decameter scale, where 0.1 to 3.9% of shear deformation was estimated to be seismic (Duboeuf et al., 2017). Similar studies in shale materials report that less than 0.1% of deformation is seismic (De Barros et al., 2016). An increased value of 4 to 8% released seismic energy was reported for hydraulic fracturing experiments on granite samples at the laboratory scale (Goodfellow et al., 2015). Also, at the field scale, large amount of aseismic deformation is suspected due to the observed slip dislocation of up to 4 cm on an acoustic televiewer log of an injection interval in granite at Soultz-sous-Forêts, which is much larger than the slip motion associated with the recorded seismic events (Cornet et al., 1997). The large amount of aseismic deformation at Grimsel is in agreement with the observation that the pressurized volume, inferred through seismic velocity changes from 4D seismic tomography, is larger than the seismically activated volume (Figure 9, b-d). Nonetheless, the seismic velocity changes above a value of -0.2% overlap well with induced seismicity fronts (Figure 9, b-d, and Doetsch et al. (2018b);Schopper et al. (under review)). Along these lines, the fault slip experiments at the LSBB underground laboratory in France, in combination with coupled hydrological and quasi-static rupture modeling, show that aseismic rupture fronts propagate faster and to greater distances compared to the diffusion of pressurized pore fluid (Guglielmi et al., 2015; Bhattacharya & Viesca, 2019). In contrast, in our case the inferred seismic velocity changes indicate that the pressurized volume may be a better proxy for the stimulated volume undergoing both aseismic and seismic deformation.

However, it should not be forgotten that when estimating the ratio of seismic to aseismic deformation or the proportion of seismically to aseismically stimulated volume, there is a spatially varying seismic network performance (see section 3.7). In section 3.6 we estimated the ratio of seismic to total deformation and counteracted the influence of the varying network performance by extrapolating seismic rates to a common minimum magnitude (Figure 12). We observed a convergence of the ratios between seismic and total deformation.

Finally, since stimulation may be to a large extent aseismic, we are left to clarify whether the locations of transmissivity increase coincide with seismic event locations or if transmissivity also increases in aseismic regions. When comparing observed seismicity and the corresponding injectivity changes in the Grimsel stimulation experiments performed on the S1 and the S3 shear zones, it becomes obvious that an increased seismic response does not necessarily represent locations of increased transmissivity. However, the Grimsel ISC project could further contribute to answering this question in future studies, because strain built up in a direct and poroelastic fashion, as well as pressure build up and pressure breakthroughs, are observed at discrete locations throughout the experimental volume before, during and after the performed stimulation experiments.

### 5.4 Outlook

The scope of future work is to gain more insight into seismic source mechanisms through moment tensor inversion of the seismic events that exhibit an adequate number of arrivals with good spatial coverage. For moment tensor inversion we intend to use the software HybridMT (Kwiatek et al., 2016). For more complete catalogs and more reliable magnitude estimates, we aim to use the matched filter technique and the relative magnitude computation approach introduced by Herrman et al. (2019). To investigate the spatially varying network sensitivity in 3D, we

intend, for the volume sampled by induced seismicity, to perform a spatial probability-based completeness study (Schorlemmer & Woessner, 2008). At locations where no seismicity was observed, a Bayesian approach is planned (Mignan et al., 2011). The new insights will be combined with strain-, tilt and pressure observations in the experimental volume and lead to a better understanding of how the rock mass responds to high-pressure fluid injection.

## 6 Implications for managing induced seismicity risk

Seismic risk management is a key requirement for the sustainable development of deep geo-energy, such as EGS (Grigoli et al., 2017; Trutnevyte & Wiemer, 2017; Lee et al., 2019). In the following, we propose potential implications for induced seismic risk management from our GTS experiments:

**Anticipate variability:** Despite comparable injection strategies and injection intervals being located with a few tens of meters, the seismic response in terms of productivity and size distribution is surprisingly variable (e.g., Figures 6, 11, 14). While an explanation for such variability may be found in retrospect, forecasting the expected seismic hazard during future injections at the GTS could be affected by large uncertainties. Thus, large uncertainties in seismic hazard forecasts for less well-known, well characterised and well monitored sites have to be anticipated. However, at the same time, the seismic response during stimulations are often surprisingly well predictable using an injected fluid volume once an estimate of the site-specific time-invariable seismogenic index is available (Mignan et al. (2017)). Possibly, the variability in the seismic response, as we observed it at the GTS, would be unified, once multiple faults in a larger region are stimulated. However, our observations suggest that the seismic response would not be an average response, but rather represent the one with the most seismogenic structures in the stimulated volume.

**Update induced seismic hazard forecasting:** Since a-priori estimates of the seismic response of a stimulation is difficult, improved forecasts with more confidence in the expected seismicity may be done after initial testing. Figure 5 illustrates that, based on the first 200 l of injected volume, it is possible to roughly forecast the overall productivity. While these forecasting strategies will need to be formally tested (e.g. following the approaches of Király-Proag et al. (2016);Broccardo et al. (2017);Király-Proag et al. (2017)), it suggests that the strategies used for adaptive traffic light systems (e.g., Grigoli et al. (2017);Mignan et al. (2017) are required and can be successful. This is also in line with the recommendation of the Pohang investigation (Lee et al., 2019).

**Injection strategies:** Our study shows the pronounced influence of geology on induced seismicity during high-pressure fluid injection. It may be possible that alternative injection schemes could have a similar pronounced impact on the seismic response but this has yet to be proven. Our results clearly suggest that great care is necessary when evaluating different injection schemes, as even within the same geological unit, the rock architecture has a pronounced influence, which raises the questions of whether it is possible to find two or more sites within an in-situ experiment that are similar enough to neglect the influence of geology and concentrate solely on the influence of different injection protocols.

**Selective stimulation (zonal isolation):** The Grimsel results recommend the concept of zonal isolation (i.e. the selective stimulation of short borehole sections). In an open hole stimulation, most injected fluid may have only

entered the most transmissive shear zones and increased their transmissivity marginally, but at the cost of an increased seismic response. From our experiment, we conclude that not only should a single pre-stimulation test per site be performed, but also a pre-stimulation in each isolated zone. Such pre-stimulations with small fluid volumes would not only allow estimation of the initial hydraulic properties, but also provide a learning phase for seismicity forecasting models. Furthermore, they not only identify structures with an increased seismic response, but also less seismogenic structures that have a larger propensity for aseismic slip. As a consequence, one should be able to skip and seal isolated zones where an increased seismic response or the chance of hydraulic short-circuits are anticipated, and focus stimulation in less seismogenic zones. However, the feasibility of zonal isolation techniques in the context of EGS have yet to be tested.

## Data availability

The Grimsel ISC Experiment Description is available at https://doi.org/10.3929/ethz-b-000310581 The seismic dataset, as well as hydraulic data of the Grimsel ISC hydraulic shearing and hydraulic fracturing experiments can be found at https://doi.org/10.3929/ethz-b-000276170 (http://hdl.handle.net/20.500.11850/280357).

## Competing interests

The authors declare that they have no conflict of interest.

## Acknowledgements

This study is part of the In-situ Stimulation and Circulation (ISC) project established by the Swiss Competence Center for Energy Research - Supply of Electricity (SCCER-SoE) with the support of Innosuisse. Funding for the ISC project was provided by the ETH Foundation with grants from Shell and EWZ and by the Swiss Federal Office of Energy through a P&D grant. Linus Villiger is supported by grant ETH-35 16-1; Hannes Krietsch is supported by SNF grant 200021_169178; Nathan Dutler is supported by SNF grant 200021_165677. The Grimsel Test Site is operated by Nagra, the National Cooperative for the Disposal of Radioactive Waste. We are indebted to Nagra for hosting the ISC project in their facility and to the Nagra technical staff for onsite support.

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
