# Peer review of "Influence of reservoir geology on seismic response during decameter scale hydraulic stimulations in crystalline rock"

_Solid Earth, 2019_

## Referee Comment (RC1) · Anonymous Referee #1 · 29 Nov 2019

The manuscript describes the in-situ injection experiments at the Grimsel Underground laboratory in detail with emphasis on the seismic response and its relation the the fracturing treatments. In particular, the authors compare the seismic responses of hydroshear and hydraulic fracturing treatments. The subject is introduced very comprehensively and with attention to detail, the paper is very well written, the figures are of great quality and all analysis steps are well thought out and executed. Hence, I only have some minor comments for improvements of the paper.

The biggest question I have left after reading the manuscript is concerned with the (lack of) difference of the HS and HF stimulation responses. It seems the experiment was set

up to tease out the differences in seismic behavior between the two different stimulation treatments. Looking at the structures that were activated seismically I wonder if the result really were ANY different. It seems that the same structures were activated and the variation among the 4 tests of each treatment was at least as large as the variation between HF and HS treatments. Although the differences of HF and HS tests are discussed in some length, I am missing a clear statement regarding this negative finding.

To support this I would like to encourage the authors to rework Fig. 4 as the pressure information cannot be discerned from that. Please add a second axis for pressure and scale it such that it uses the full range of the subfigure.

The last major comment is regarding the references in the text, many of which are missing in the list of references. I did not do a full check, but urge the authors to do so. Some missing references are McClure&Horne, 2013; Secor & pollard, 1975; Schoenball et al., 2019; Kwiatek et al., 2018; Goodfellow & Young, 2014, Brixel et al., in review; Villiger et al., 2019; Jung, 2013b;

Minor comments: L. 115: I would not say that alternative approaches to McGarr are more conservative, but rather say that the assumptions of McGarr may not be valid. We now have ample of data to discard the McGarr hypothesis.

L. 261: It is unclear what "a possibly unperturbed stress state" should mean.

l. 537f: Do you have an idea why so many events were detected after shut-in in this test? Any indications from the structures that were active. How about b-value, etc.? Is that something we should worry about for a full-scale test?

Fig. 6: Would be helpful to remind what gray events represent

l. 655: You substitute t by the injected volume for Shapiro's diffusivity estimation (SBRC). This addresses and important criticism of the SBRC method, namely that it disregards fluid injection rate (e.g. Schoenball et al., 2010, GJI). I believe this deserves

to be discussed in some more detail here.

l. 694 & 696: -0.1%, -0.5%

l. 772ff: Not quite clear. Do you mean the moment of all events combined into a single event?

l. 956ff: Nice discussion, important observation!

l. 991ff: In this whole discussion I am missing some statements whether the activated structures actually are hydraulic fractures. Based on the Schmitt plots and the seismicity plots it seems that most structures could be hydraulic fractures just as well.

Section 5.3: Given the data presented in this manuscript this entire discussion is rather speculative. I can see that this point may well be made using complementary data discussed in some of the "in prep" manuscripts referenced throughout but it may not fit well in this manuscript.

l. 1089ff: This is an interesting discussion. However, there should be a limit to this given by the total volume per stage or the size of the activated rock volume. Let's call this elementary rock volume (ERV) for now. Once you break out of this ERV, your likelihood of activating a structure from a neighboring ERV with different seismogenic index grows very fast. So unless you are activating only the near-wellbore region of your ERV there may be limits to the zonal isolation approach.
* * *

---

## Referee Comment (RC2) · Anonymous Referee #2 · 4 Feb 2020

The manuscript titled "Influence of reservoir geology on seismic response during decimeter scale hydraulic stimulation in crystalline rock" by Villiger and others is a dense description of a series of injection experiments performed at the Grimsel Test Site, Switzerland. The experiment was well-constructed and the seismic analysis is very complete. While I find the science aspects of this paper intriguing, the organization and in some some cases level of detail distracts from the main topic. The paper seems to be structured like a report more so than a journal article. A lot of detail is included whether it is relevant to the main goal or not. Along those lines, I am not sure that the outlook section describing the researchers next plans is appropriate here. Overall, I find this an important paper with interesting and possibly significant results. However, had I not been reviewing, I probably would have quit reading, because it was hard to keep the track of the goal and relevant information given the organization and distracting information.

The abstract is nice and conveys the main points and conclusions. The section on the Study Site is also nice with a good figure. There is also good information throughout the paper. However, I would suggest looking at the organization to more clearly convey the main points. Below I provide some specific comments for each section.

**Introduction**
This section is packed full of interesting and useful information of EGS, but at 7 pages seems like a lot to introduce the study at hand, and given the organization while the information is useful and interesting there is not a clear motivation between the introduction and what is to come. The title talks about geology, but this is not introduced until page 3.

Some specific comments:
Organization of paragraph starting Line 124 needs work: line 124 "seismicity rates might be linked to geologic setting"; line 127 "Seismicity may also be dependent on fault orientation"; and then line 131 returns to "seismic response . . .may also be linked to local geological setting". Seems like should talk about local geologic setting generally and then go into details like fault orientation. As written now is jumpy and distracting.

Figure 1: what is Petrothermal? Should be defined somewhere, not a common term

**Methods**
This section describes the experimental design for injection and methods used in the catalog construction. Table 1 is a nice concise collection of both the experimental design and results from the seismic analysis. I would suggest a second table that breaks out the seismic information by cycle. You later describe cycles but there is not a clear place to assess this analysis.

The discussion of the AE sensors was really distracting. Strongly suggest moving this discussion to the supplement and just highlight the key points in a subsection "Integration of AE sensors"

Specific comments:
The pick errors seem very specific were they determined empirically? If not how did you decide on these values?

You spend a lot of time discussing pick weights and the velocity model, but when it comes to station corrections there is one sentence referring to another reference. Seems that there could be more here.

The discussion of magnitudes and notation is confusing:

The introduction of the three types of magnitudes paragraph starting line 418 is confusing. I recommend that you start with a direct sentence like "Here we calculate three magnitudes: . . ."

Line 513-514 discusses how you calculate the "adjusted amplitude magnitude" MA (as defined line 426), but in line 514 this is called the "amplitude magnitude". Please be consistent.

The equation to get the adjusted magnitude is to subtract 4 from the relative magnitude calculation. This seems really large. In the comparison how large was the spread in magnitudes. It would be worth seeing a figure of the Mr Mw comparisons

**Results**
Specific Comments:
Line 536 says "During HF injections, significantly fewer detections compared to HS injections (Figure 4c,d)". In Fig 4c, the cumulative number of seismic events is 2000 compared to Fig 4a with 500. Even Fig 4d has 600 which is more than 4a. True HF is less than what is shown in Fig 4b, but your statement is not supported by the figure. When I look at numbers in Table 1, I would still be pushed to use the word "significantly". There actually seems to be a lot of variability in the total number of detections, which should perhaps be investigated or at least commented on.

Line 537: For HF "a comparably high percentage of detections (33%) were made during shut-in". This is not evident in Figure 4.

Figure 5a would help to label which were S1 and which S3 injections

Instead of "fitted for" suggest "fit to" throughout text

Line 760, You introduce M0 displacement and M0 hydraulic and then immediately move to a discussion of seismic moment. It would be helpful to have a conceptual description of these parameters when you introduce the terms and how they differ before going into the details.

Line 801: is this really "a best guess"?

The section on Network Performance should come earlier in the discussion and be used to inform the discussion on b-value and the seismic cloud.

Line 868: Why is there a discussion of S-phases. You did not use them. You could simply add a sentence when showing the waveforms in Fig 3 that the S-phases were not of sufficient quality for picking

**Discussion, conclusions, and outlook**
Given all the information in the Results, I was looking for a concise summary that linked back to main the questions of the paper. This could be an introductory paragraph before diving into the details.

Line 880: you mention "permeability increase, pressure propagation and rock deformation". These were not directly addressed in the paper

Line 925 what does "the first 100 1 of fluid" mean?

Line 175: what does "first 200 1 of injected volume" mean

---

## Author Comment (AC1) · 3 Mar 2020

**Author's reply se-2019-159**

Dear Editor

We would first of all like to thank you and the reviewers for your valuable time you put into this manuscript. Your comments and constructive criticism have improved the quality of this manuscript. In the following, we address all reviewer comments point by point in blue color font. The reviewer comments are held in italic font. The line locations used in this reply letter are referenced to line locations with displayed track changes.

**Anonymous Referee #1**

*The biggest question I have left after reading the manuscript is concerned with the (lack of) difference of the HS and HF stimulation responses. It seems the experiment was set up to tease out the differences in seismic behavior between the two different stimulation treatments. Looking at the structures that were activated seismically I wonder if the result really were ANY different. It seems that the same structures were activated and the variation among the 4 tests of each treatment was at least as large as the variation between HF and HS treatments. Although the differences of HF and HS tests are discussed in some length, I am missing a clear statement regarding this negative finding.*

We thank the referee for this observation and agree that our experiments were also set up to tease the differences between HF and HS stimulation experiments. The differences may be less than expected, because HFs quickly connect to pre-existing fractures. We added a sentence explaining this assumption in the discussion section 5.2 (L. 1139). However, we also believe that several clear and detailed statements on the differences were already included in the first section of the discussion part (starting in L. 1036) as well as in the abstract (L. 30) of the manuscript.

*To support this, I would like to encourage the authors to rework Fig. 4 as the pressure information cannot be discerned from that. Please add a second axis for pressure and scale it such that it uses the full range of the subfigure.*

Yes, we agree with the referee, and in order to better discern the injection pressure from the injection rate an additional y-axis was added to Figure 4 and to the figures of the remaining experiments in the supplementary material as the referee suggests.

*The last major comment is regarding the references in the text, many of which are missing in the list of references. I did not do a full check, but urge the authors to do so. Some missing references are McClure&Horne, 2013; Secor & pollard, 1975;Schoenball et al., 2019; Kwiatek et al., 2018; Goodfellow & Young, 2014, Brixel et al., in review; Villiger et al., 2019; Jung, 2013b;*

We regret this mistake. A full check was performed and missing references were added to the reference list.

*Minor comments:*

*L. 115: I would not say that alternative approaches to McGarr are more conservative, but rather say that the assumptions of McGarr may not be valid. We now have ample of data to discard the McGarr hypothesis.*

Yes, we agree with the referee here. However, to shorten the manuscript for better readability we removed the part on estimating the maximum possible magnitude to a great extend, because our paper does not contribute to this discussion.

*L. 261: It is unclear what "a possibly unperturbed stress state" should mean.*

Yes, this can be confusing. We now added more information to the text to make clear what we mean by the unperturbed and the perturbed stress state (starting in L.315).

*L. 537: Do you have an idea why so many events were detected after shut-in in this test? Any indications from the structures that were active. How about b-value, etc.? Is that something we should worry about for a full-scale test?*

That is an interesting question! Unfortunately, we believe this high percentage of detections (33%) has no seismic origin. We think a direct hydraulic connection between the injection interval and the open seismic monitoring boreholes (termed GEO's, Figure 3) was created. The hydraulic connection led to flow-through of the seismic monitoring boreholes during stimulation and possibly to stick-slip movements of the AE sensors, which in turn led these detections (explanation starting in L. 604). Further evidence that these detected events were induced through flow-through provides the fact that these detections were often made on AE sensor pairs placed in the same seismic monitoring borehole. We remember that a detection is declared a detection as soon as a possible seismic event was observed on at least two AE sensors (L. 411).

*Fig. 6: Would be helpful to remind what gray events represent*

Yes, agreed, a sentence was added to the caption of Figure 6 to clarify what the gray events are.

*L. 655: You substitute t by the injected volume for Shapiro's diffusivity estimation (SBRC). This addresses and important criticism of the SBRC method, namely that it disregards fluid injection rate (e.g. Schoenball et al., 2010, GJI). I believe this deserves to be discussed in some more detail here.*

We absolutely agree and added a statement that the estimated diffusivity values are based on the SBRC which disregards the fluid injection rate (L. 733).

*L. 694 & 696: -0.1%, -0.5%*

To direct the reader more to the main topic of the manuscript and improve the readability we decided to delete Figure 9 in which we superimposed seismicity with velocity variations.

*L. 772ff: Not quite clear. Do you mean the moment of all events combined into a single event?*

Yes, exactly. We made adjustments to the text, the sentence now reads "…cumulating the moment release of all possible seismic events per injection experiment into a single earthquake would have induced a moment magnitude $M_W$ in the range of -3 to -1" (starting in L. 884).

*L. 956ff: Nice discussion, important observation!*

Thanks!

*L. 991ff: In this whole discussion I am missing some statements whether the activated structures actually are hydraulic fractures. Based on the Schmitt plots and the seismicity plots it seems that most structures could be hydraulic fractures just as well.*

We agree with the referee; these statements are missing. We added the information on what our definition of a hydraulic fracture is and assigned the probable experiments in which a hydraulic fracture was induced to it. We furthermore added the information to the text, that the combination of mode-I and mode-II,III deformation is possible for the majority of the experiment (starting L. 1120).

75

*Section 5.3: Given the data presented in this manuscript this entire discussion is rather speculative. I can see that this point may well be made using complementary data discussed in some of the "in prep" manuscripts referenced throughout but it may not fit well in this manuscript.*

80  We fully agree with the referee. We now just state the determined percentages of seismic to total deformation observed in our experiments, compare it to values form the literature and deleted the rest of the paragraph (starting in L. 1160).

*L. 1089ff: This is an interesting discussion. However, there should be a limit to this given by the total volume per stage or the size of the activated rock volume. Let's call this elementary rock volume (ERV) for now. Once you*
85  *break out of this ERV, your likelihood of activating a structure from a neighboring ERV with different seismogenic index grows very fast. So unless you are activating only the near-wellbore region of your ERV there may be limits to the zonal isolation approach.*

Yes, we believe the referee is absolutely right here. We pose the question on how representative a pre-stimulation might be in the text. We conclude that zonal insulation and the ability to seal isolated zones no matter what offers
90  more flexibility and opportunities to intervene an ongoing stimulation treatment (starting in L. 1237).

---

## Author Comment (AC2) · 3 Mar 2020

**Author's reply se-2019-159**

Dear Editor

We would first of all like to thank you and the reviewers for your valuable time you put into this manuscript. Your comments and constructive criticism have improved the quality of this manuscript. In the following, we address all reviewer comments point by point in blue color font. The reviewer comments are held in italic font. The line locations used in this reply letter are referenced to line locations with displayed track changes.

**Anonymous Referee #2**

**Opening paragraph**

*The manuscript titled "Influence of reservoir geology on seismic response during decimeter scale hydraulic stimulation in crystalline rock" by Villiger and others is a dense description of a series of injection experiments performed at the Grimsel Test Site, Switzerland. The experiment was well-constructed and the seismic analysis is very complete. While I find the science aspects of this paper intriguing, the organization and in some some cases level of detail distracts from the main topic. The paper seems to be structured like a report more so than a journal article. A lot of detail is included whether it is relevant to the main goal or not.*

We thank the referee for this criticism. We realize that the manuscript/report may be too bulky. We shortened and streamlined it to be more concise and readable and devoid of distracting details.

Aside of a substantial shortening of the introduction, we merged section "4.2 Spatial properties of seismicity clouds" and a shortened section "4.3 Propagation of seismicity". Left of section 4.3 are diffusivity values and general propagation characteristics of seismicity clouds, whereby the methodological part of these sections was transferred to the supplementary material SM6 and SM5, respectively. In addition, Figure 9, where we showed seismic triggering fronts of an experiment, as well as an overlay of seismicity with inferred velocity changes, was removed. Also, section "4.7 Network performance" was removed partially from the manuscript. From Figure 13 only d) is left and included in section "4.3 Frequency magnitude distributions". Finally, parts of the discussion in section "5.3 Aseismic deformation" and the outlook section was removed from the manuscript.

*Along those lines, I am not sure that the outlook section describing the researchers next plans is appropriate here.*

We removed the outlook section as recommended.

*Overall, I find this an important paper with interesting and possibly significant results. However, had I not been reviewing, I probably would have quit reading, because it was hard to keep the track of the goal and relevant information given the organization and distracting information.*

**Introduction**

*like a lot to introduce the study at hand, and given the organization while the information is useful and interesting there is not a clear motivation between the introduction and what is to come. The title talks about geology, but this is not introduced until page 3.*

We agree with the referee that the introduction covered a wider scope as the rest of the article. We shortened and restructured the introduction so that it is focussed to the main topics of the paper.

40  *Some specific comments: Organization of paragraph starting Line 124 needs work: line 124 "seismicity rates might be linked to geologic setting"; line 127 "Seismicity may also be dependent on fault orientation"; and then line 131 returns to "seismic response . . .may also be linked to local geological setting". Seems like should talk about local geologic setting generally and then go into details like fault orientation. As written now is jumpy and distracting.*

45  This whole paragraph was restructured and rewritten (now starting in L. 121).

*Figure 1: what is Petrothermal? Should be defined somewhere, not a common term*

Agreed, Petrothermal (i.e., injections into hot and dry rock volumes) is now defined in the caption of Figure 1 as well as Hydrothermal (i.e., injection into aquifers).

***Methods***

50  *This section describes the experimental design for injection and methods used in the catalog construction. Table 1 is a nice concise collection of both the experimental design and results from the seismic analysis. I would suggest a second table that breaks out the seismic information by cycle. You later describe cycles but there is not a clear place to assess this analysis.*

We understand the referee comment. We included tables in which located seismic events of HS and HF experiments are resolved in cycles and phases to the supplementary material SM8. We believe the tables are cumbersome to read and therefore placed them in the supplementary material.

*The discussion of the AE sensors was really distracting. Strongly suggest moving this discussion to the supplement and just highlight the key points in a subsection "Integration of AE sensors"*

Thanks for this comment. We assume that the referee is talking about section "3.2.1. Seismic monitoring" and the explanation of the installed hardware therein. We do not agree with the referee here. This is not the usual setup of a seismic network, therefore, we belief it is important to assign it some relevance and explain it in some detail (Starting in L. 369).

*Specific comments: The pick errors seem very specific were they determined empirically? If not how did you decide on these values?*

65  Yes, pick uncertainties were determined empirically. We added the word "empirically" at the location where we introduce the P-wave pick uncertainties (L. 432).

*You spend a lot of time discussing pick weights and the velocity model, but when it comes to station corrections there is one sentence referring to another reference. Seems that there could be more here.*

We agree with the referee, that the Joint Hypocenter Determination (JHD) approach which involves the determination of station corrections is not discussed in much detail. We have added one more sentences to the explanation of the station corrections. However, the weighting of the P-wave picks in the location of the seismic events, as well as the determination of the five velocity parameters using a genetic algorithm are introduced in this manuscript. The applied station corrections on the other hand follow the approach introduced by Gischig et al. (2018). Gischig et al. (2018) offer a nice summary in their appendix of the Joint Hypocenter Determination (JHD) approach

75 in an anisotropic velocity model, in which the determination of station corrections is a central part. Therefore, we decided to not explain the station corrections in more detail (L. 460).

*The discussion of magnitudes and notation is confusing: The introduction of the three types of magnitudes paragraph starting line 418 is confusing. I recommend that you start with a direct sentence like "Here we calculate three magnitudes: . . ."*

80 Yes, we agree with the referee here, this can be confusing. We inserted some introductory sentences in the direction the referee proposes (starting L. 477).

*Line 513-514 discusses how you calculate the "adjusted amplitude magnitude" MA (as defined line 426), but in line 514 this is called the "amplitude magnitude". Please be consistent.*

Yes, that is confusing, now $M_A$ is called the "amplitude magnitude" throughout the manuscript (L. 480, 493,
85 582).

*The equation to get the adjusted magnitude is to subtract 4 from the relative magnitude calculation. This seems really large. In the comparison how large was the spread in magnitudes. It would be worth seeing a figure of the Mr Mw comparisons*

In our understanding the adjustment from relative magnitudes $M_r$ which aims to describe the relative source
90 strength with no absolute meaning, into more realistic amplitude magnitude $M_A$ based on a physics based $M_W$ is arbitrary. We included a moment magnitude $M_W$ vs. amplitude magnitude $M_A$ comparison to the supplementary material SM7.

**Results**

95 *Specific Comments: Line 536 says "During HF injections, significantly fewer detections compared to HS injections (Figure 4c,d)". In Fig 4c, the cumulative number of seismic events is 2000 compared to Fig 4a with 500. Even Fig 4d has 600 which is more than 4a. True HF is less than what is shown in Fig 4b, but your statement is not supported by the figure. When I look at numbers in Table 1, I would still be pushed to use the word "significantly". There actually seems to be a lot of variability in the total*
100 *number of detections, which should perhaps be investigated or at least commented on.*

Yes, we agree with the referee, this may be confusing. Detected seismic events do not have to be mistaken with located seismic events. Seismic detections should only in a first approximation be considered as a measure for seismic activity. Seismic detections can be flawed because the quality control of location (i.e., at minimum 4 P-wave arrivals are needed, and the largest axis of the error ellipsoid should be within 1.5m) has not yet been passed.
105 One reason why the number of detections can be flawed is the assumed flow through a seismic monitoring borehole and the triggering of stick-slip movement of AE sensors in the borehole (starting L. 608). For any analysis in this manuscript only located seismic events are used.
To make the difference between detected and located events more clear, the seismic detection rate was removed from the middle plots of Fig. 4 and the cumulative located events are shown on the second y-axis of each experi-
110 ment.

*Line 537: For HF "a comparably high percentage of detections (33%) were made during shut-in". This is not evident in Figure 4.*

We also agree with the referee here; this is not obvious from Figure 4. Figure 4 only hosts the time evolution of a selection of HF experiments, to get a full picture, the supplementary material SM3 has to be considered. To the cross reference of the figure we added "for a selection of HF experiments" and "for a selection of HS experiments" for HF and HS experiments, respectively (L. 605 and L. 589).

Contributions to the 33% shut-in detections stem from experiment HF5 and HF8 in which a hydraulic connection was created to a seismic monitoring borehole (explanation starting in L. 608). We additionally indicated the period where we believe the detections are flawed by stick-slip movements of the sensors in the monitoring borehole (figure to HF5 in SM3, and figure to HF8 is Figure 4d).

*Figure 5a would help to label which were S1 and which S3 injections*

Yes, that is right! The information was added to the figure.

*Instead of "fitted for" suggest "fit to" throughout text*

That does fit better. In L. 687 the "fitted for" was exchanged by "fitted to".

*Line 760, You introduce M0 displacement and M0 hydraulic and then immediately move to a discussion of seismic moment. It would be helpful to have a conceptual description of these parameters when you introduce the terms and how they differ before going into the details.*

The three quantities (seismic moment, hydraulic moment and total moment release) used in this section are introduced in an introductory sentence. Then, the estimate of the seismic moment release is explained in more detail (starting in L. 873 onwards). In a next section (L. 890) the quantity equivalent hydraulic moment and its estimation is explained. Finally, the total moment (L. 898) is explained. We feel the structure will be clear to readers when the paragraph is read as a whole.

*Line 801: is this really "a best guess"?*

We agree with the referee here that "a best guess" is maybe not the best choice of wording here. We changed "a best guess" to "average estimate" (starting L. 879).

*The section on Network Performance should come earlier in the discussion and be used to inform the discussion on b-value and the seismic cloud.*

Thank you for this comment. We agree with the referee and included the discussion on a varying network sensitivity at the location where $M_C$ is introduced (Section 4.3, L. 808). We also inserted a new figure showing the varying $M_C$ in the experimental volume (Figure 9).

*Line 868: Why is there a discussion of S-phases. You did not use them. You could simply add a sentence when the waveforms in Fig 3 that the S-phases were not of sufficient quality for picking*

We agree with the referee, it is not common sense to discuss this limitation. However, we believe when displaying waveforms in Figure 3, some readers might ultimately ask themselves why no S-phases are observed and picked. Thus, we added some more information at the location where we explain that no S-waves were picked (starting L. 422).

***Discussion, conclusions, and outlook***

*Given all the information in the Results, I was looking for a concise summary that linked back to main the questions of the paper. This could be an introductory paragraph before diving into the details.*

150    Yes, we totally agree with the referee. An introductory paragraph was added at the beginning of this section summarizing the main results of this experimental campaign out of seismological perspective (starting in L. 1002).

*Line 880: you mention "permeability increase, pressure propagation and rock deformation". These were not directly addressed in the paper*

Yes, we agree, in this section we refer to the performed hydraulic stimulation experiments at the Grimsel Test Site

155    in general. It is meant as introduction as the discussion brings together other observations made prior and during the experiments such as injectivity change or the influence of geology (L. 992).

*Line 925 what does "the first 100 1 of fluid" mean?*

Yes, we understand that this can be confusing. We mean by the first 100 l, the initial 100 l of fluid which were injected into the ductile shear zones S1. We replaced "first" with "initial" leading to "the initial 100 l of fluid"

160    (now L. 1054).

*Line 1077: what does "first 200 1 of injected volume" mean*

Same here, we replaced "first" with "initial" (now L. 1214).

---

## Author Response (AR2)

**Author's reply se-2019-159**

Dear Michael Malinowski (Topical Editor)

We thank you for editing our manuscript, which we believe has improved significantly in quality throughout the course of this review process! In the following, we address your comment, held in italic font, in blue color font. The line location used in this reply letter refers to the line location with displayed track changes.

**Topical Editors comment**

*Thank you for the revised manuscript. I believe it benefited greatly from the reviewers' comments, especially as some parts were shortened now. However, before final acceptance, I urge you to consider rewriting the last part of the Introduction section, starting with "The specific objectives for this publication were to...". To me, it sounds like general project goals. I suggest removing it or replacing it with a "walk-through" the paper: First, we show... Then we do... Finally, we show...*

We agree with the editor here and rewrote the last paragraph of the introduction. The paragraph now gives a quick walk through the paper as the editor suggested (starting in L. 121).

[revised manuscript text omitted]